# Systematic tissue annotations of genomics samples by modeling unstructured metadata

Nathaniel T. Hawkins[1], Marc Maldaver [1], Anna Yannakopoulos[1], Lindsay A. Guare [1,2,3] & Arjun Krishnan [1,2,4] ✉

There are currently >1.3 million human −omics samples that are publicly available. This valuable resource remains acutely underused because discovering particular samples from this ever-growing data collection remains a significant challenge. The major impediment is that sample attributes are routinely described using varied terminologies written in unstructured natural language. We propose a natural-language-processing-based machine learning approach (NLP-ML) to infer tissue and cell-type annotations for genomics samples based only on their free-text metadata. NLP-ML works by creating numerical representations of sample descriptions and using these representations as features in a supervised learning classifier that predicts tissue/cell-type terms. Our approach significantly outperforms an advanced graph-based reasoning annotation method (MetaSRA) and a baseline exact string matching method (TAGGER). Model similarities between related tissues demonstrate that NLP-ML models capture biologically-meaningful signals in text. Additionally, these models correctly classify tissue-associated biological processes and diseases based on their text descriptions alone. NLP-ML models are nearly as accurate as models based on gene-expression profiles in predicting sample tissue annotations but have the distinct capability to classify samples irrespective of the genomics experiment type based on their text metadata. Python NLP-ML prediction code and trained tissue models are available at https://github.com/krishnanlab/txt2onto.

Currently, there are data from >1.3 million human−omics samples and >26,000−omics datasets that are publicly available in repositories like the EBI ArrayExpress[1,2] and NCBI Gene Expression Omnibus[3] (GEO). These samples capture cellular responses of diverse human tissues/cell types under thousands of conditions, making these published data invaluable for other researchers to reuse for various tasks: (i) reanalyze them to answer new questions, (ii) check reproducibility of original findings, iii) perform integrative-/meta-analysis across multiple studies, (iv) review earlier studies for support of new data/findings, and (v) generate large-scale data-driven hypotheses for experimental follow-up. However, these data remain acutely underused because discovering all the samples relevant to one's interest, quickly and thoroughly, from this ocean of data is still a major challenge. This is because samples are routinely described using non-standard, varied terminologies written in unstructured natural language even though guidelines for describing data submitted to public repositories were established as early as 2001[4].

To address this issue, considerable effort has been invested in creating sample metadata hubs like Biosamples[5] where human curators use semi-automated workflows[6] to manually annotate samples by

[1]Department of Computational Mathematics, Science and Engineering, Michigan State University, East Lansing, MI 48824, USA. [2]Department of Biochemistry and Molecular Biology, Michigan State University, East Lansing, MI 48824, USA. [3]Department of Microbiology and Molecular Genetics, Michigan State University, East Lansing, MI 48824, USA. [4]Department of Biomedical Informatics, University of Colorado Anschutz Medical Campus, Aurora, CO 80045, USA. ✉e-mail: arjun.krishnan@cuanschutz.edu

assigning them with terms in controlled vocabularies or ontologies[7] such as the Uber-anatomy ontology[8] (Uberon). Others have developed similar web platforms for manual sample annotation[9] and explored crowdsourcing to curate sample annotations[10,11]. The latter efforts are usually case studies and rely on an interested and available population of individuals to spread out the burden of manual curation. While curation efforts often lead to high-quality gold standards, automated high-throughput methods are needed to keep up with the scale of available samples, which already totals to more than a million and continues to grow exponentially[12–15].

To meet this challenge, a number of methods have recently been developed that use natural language processing to computationally infer standardized sample annotations from their text descriptions[16]. Most of these methods are based on biological 'named entity recognition' (NER), the task of automatically recognizing words or phrases in plain text that correspond to known biological entities such as tissues, phenotypes, and species, uniquely identified using terms in various ontologies. NER has been applied by searching for common phrases in experiment descriptions followed by finding matches to terms in a controlled vocabulary[17] or by generating all possible $n$-grams from experiment descriptions before finding term matches[18]. These approaches heavily rely on exact text matching and are not robust to misspellings and acronyms or abbreviations. Tools such as Metamap[19] and ConceptMapper[20] help deal with spelling variants, abbreviations/acronyms, and synonyms, and have been applied to annotate ChIP-seq data[21]. Others have used NER-based heuristic searches, including the use of regular expressions, to achieve similar outcomes[10,22]. However, without an additional step of manual curation, NER-based methods suffer from high false-positive rates due to the presence of varied and conflicting pieces of information in sample descriptions. For instance, tissue annotation methods need to deal with information about other aspects of the sample and experiment, including mentions of more than one tissue/cell type.

Newer studies have helped overcome some of these challenges, improving on NER by incorporating some structured information available in sample metadata entries in the form of key-value pairs. For example, checking specific metadata entries against predefined rules about word/phrase context has been shown to be helpful in inferring technology platform, sample type, organism, molecule type, and labeling compound[23]. A method called MetaSRA[24] also significantly improves upon NER by examining key-value pairs using graph-based reasoning over the structure of existing ontologies. The annotation scheme in MetaSRA is robust to misspellings and acronyms or abbreviations. Additionally, it can check for logical consistencies in the knowledge graph, which ensures high-quality annotations. However, MetaSRA software is currently written in Python 2, which is no longer officially supported. Further, we observe that this complex method has long runtimes and still incurs a number of false positives (see Results). Finally, some attributes about a sample might be missing in the input text but would be identifiable by association (e.g., mentioning 'cerebral palsy' while omitting source tissue 'muscle'). Some recent studies are finding that creating numerical representations of parts of sample text descriptions can help with this problem. Specifically, NER using representations created by a recurrent neural network trained on key-value pair metadata has been shown to result in models that can predict if short segments of study titles correspond to any one of 11 sample attributes[25]. Here, tissues and cell types are among the attributes that are hard to recognize. Previously, term-frequency-based vectors and topic modeling have been shown to be useful in creating such representations to inform general metadata prediction[26], which does not include standardized annotations of tissues and cell types.

We have developed a scalable approach, NLP-ML, that combines natural language processing (NLP) and machine learning (ML) to annotate samples to their tissue-of-origin solely based on their unstructured text descriptions. In the following sections, we first describe the details of NLP-ML and then demonstrate that NLP-ML outperforms exact-string matching and MetaSRA (that uses graph-based reasoning) in inferring tissue annotations from sample metadata. We then explore the benefits and limitations of supplementing predictions based on sample descriptions with those based on the description of the entire dataset. We observe that NLP-ML models of anatomically related tissues have similar model coefficients. By applying these models to classify the descriptions of biological processes and diseases, we demonstrate that NLP-ML models capture general, biologically meaningful text signals. The final sections highlight how NLP-ML models are as accurate as models trained on the sample's expression profile in classifying transcriptome samples and can, additionally, classify samples from many other experiment types using the common currency of text descriptions.

## Results

### Overview of NLP-ML for sample tissue annotation

The central idea behind NLP-ML is that unstructured sample descriptions can be represented as numerical embeddings that can serve as input to supervised machine learning classifiers to accurately predict the tissue-of-origin of samples (Fig. 1).

The input to our method is unstructured sample descriptions where each description is a bag-of-words consisting of values in all metadata fields for that sample. Considering metadata as a bag-of-words allows our approach to handle the immense variability between the types of experiments and the quality and completeness of researcher-submitted metadata. For instance, one cannot guarantee that the same metadata fields will be available for hundreds of thousands of samples. We then preprocess each of these sample descriptions to remove punctuation, remove words that contain numeric characters, lemmatize to cast words to their root forms, and finally turn all words to lowercase.

Using the preprocessed words associated with each sample description, we then create a numerical representation for that sample description using pre-trained embedding models from the flair library[27], trained on a general English and a biomedical text corpus. Flair creates word embeddings by examining character-level information. This makes the associated representations robust to misspellings and allows us to create word embeddings for words that did not appear in the corpus the original neural network models were trained on. The embedding for the whole sample description is calculated by averaging the embeddings of the constituent words, weighted by the inverse document frequency of each word estimated based on the PubMed text corpus (see Methods). Though this weighting did not show impact on performance (Fig. S1), we decided to use weighted averaging based on the hypothesis that sample embeddings influenced by infrequent words could lead to more interpretable models.

These description-based sample embeddings can serve as features in supervised machine learning models that are trained to predict the sample's tissue-of-origin. To train such models, we constructed a gold-standard containing tissue labels for about 11,600 human gene-expression samples based on annotations from a previous study[28] that we mapped to and propagated based on the UBERON-CL tissue and cell ontology (see Methods; Table 1). In total, our gold standard contains annotations for 81 unique tissues and cell types from over 300 datasets. After expanding these annotations and filtering for high-information content terms (see Methods), we were able to train models for 153 tissues and cell types. The median number of 'positive' samples for each tissue/cell-type model was ~50. Negative samples were chosen in a way consistent with the term relationships in UBERON-CL (see Methods).

Then, using this gold standard, we trained a series of one-vs-rest L1-regularized logistic regression models—one model per tissue term—using the sample embeddings as features to separate samples from a

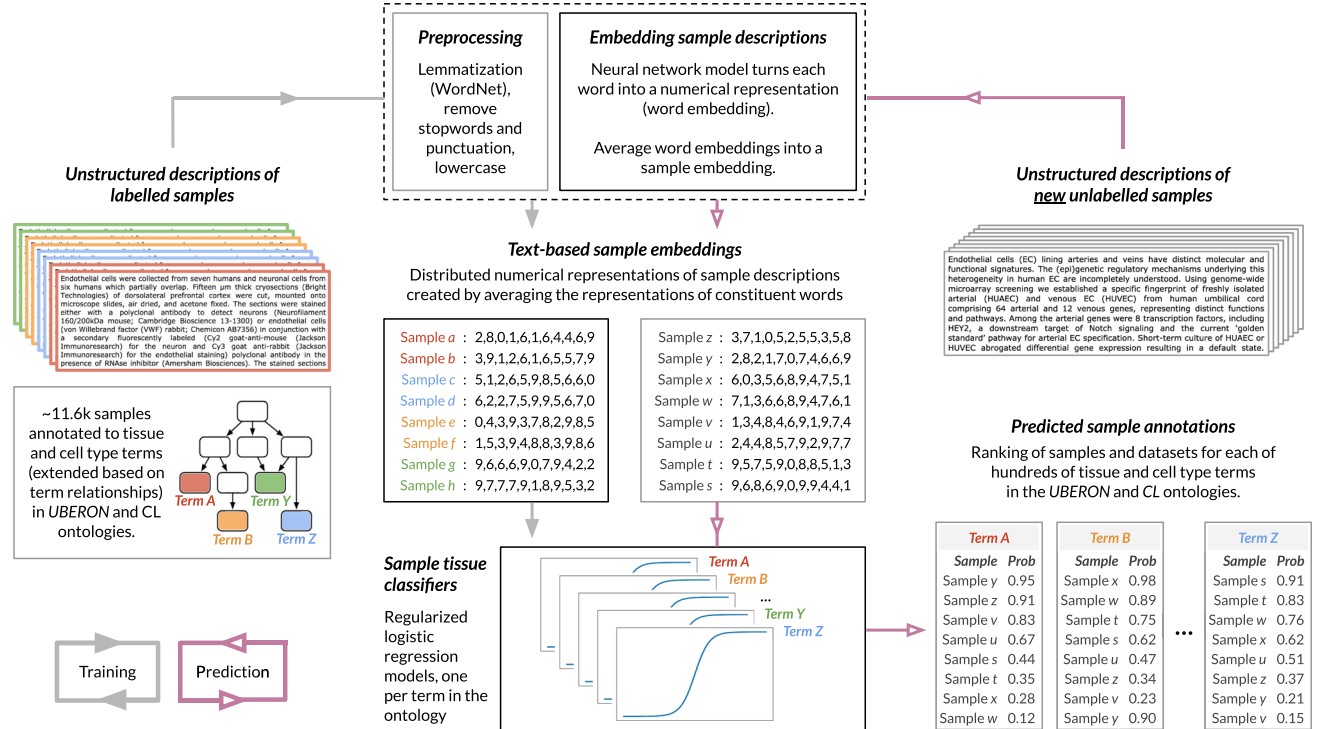

**Fig. 1 | Overview of NLP-ML.** The NLP-ML approach contains four steps: (i) Text preprocessing: Unstructured metadata of samples are preprocessed to remove text elements extraneous to sample classification and reduce words to their roots; (ii) Creating text-based sample embeddings: A neural network model trained on large text corpora is used to create numerical embeddings of individual words. An embedding of a sample is created by averaging the embeddings of the words in that sample's metadata; (iii) Training sample tissue classifiers: Supervised machine learning models—one per tissue/cell type—are trained using sample text embeddings as features and manually curated sample to tissue/cell-type annotations as labels; and (iv) Classifying new samples: Descriptions of unlabeled samples are preprocessed and turned into numerical embeddings. Each trained model takes these embeddings as input and provides the probability that the sample is from that tissue/cell type.

particular tissue from samples from other unrelated tissues. We tested multiple classifiers and observed that support vector machines and logistic regression performed the best (Fig. S2). We picked L1-regularized logistic regression because it results in sparse models that are more likely to be generalizable to new text, and naturally return probabilities at the prediction stage. Finally, tissue predictions can be made for new sample descriptions—or, in fact, any snippet of text of interest—by preprocessing the text, creating an embedding using a weighted average of word embeddings for tokens in the text, and then running the text embeddings through each of our pre-trained tissue models to get a predicted probability that the given piece of text can be annotated for each one of the tissues or cell types.

**Comparison of text-based methods for sample annotation**

NLP-ML relies solely on free-text (i.e., unstructured) metadata. To evaluate its effectiveness, we compared it to two other methods that

similarly associate text with tissue and cell-type labels. These two methods are representative of the two most common approaches in the literature. TAGGER[29] is a named-entity recognition (NER) method that looks for exact-string matches of words/phrases in a dictionary in the input text. MetaSRA[24] uses sophisticated graph-based reasoning and prior knowledge in the form of ontologies to do an NER-like task. Figure 2 shows the area under the precision-recall curve (auPRC) values for NLP-ML models along with those of TAGGER and MetaSRA based on cross-validation (CV). In these CV runs, samples from the same dataset were kept together (i.e., never split across training and testing) to avoid overestimation of prediction performance due to experiment/batch effects (see Methods).

First, we observe that our NLP-ML models significantly outperforms the other two text-based methods (higher auPRC for 74% of terms, Wilcoxon $p$-value = 2.01e-5 against MetaSRA; higher auPRC for 86% of terms, $p$-value = 2.27e-27 against TAGGER, Fig. S3). NLP-ML has a median auPRC of 0.74, which means that, out of the top 100 predicted samples for a given tissue, on average, 74 of them are correct. Not surprisingly, TAGGER consistently performs the worst due to incurring a large number of false negatives by only looking for exact-string matches in order to make an annotation. MetaSRA, with its careful processing of key and value data in the description based on ontologies, achieves a median auPRC of 0.48 and significantly outperformas TAGGER (higher auPRC for 69% of terms, Wilcoxon $p$-value = 1.12e-13).

While the median performance of NLP-ML is substantially higher than that of MetaSRA, the boxplots also indicate that there are a number of tissues for which MetaSRA performs better. To better understand the differences in relative performance of NLP-ML and MetaSRA, we first summarized method performance for tissues within

**Table 1 | Summary of tissue/cell-type gold-standard size and attributes of the associated sample and dataset metadata**

| | Median | Range |
|---|---|---|
| No. samples per dataset | 16 | 1–525 |
| No. samples per tissue/cell type | 48 | 1–1590 |
| No. samples per tissue/cell type after propagation | 121 | 1–3683 |
| No. tissue/cell-type annotations per sample after propagation | 5 | 1–39 |
| Lengths of sample descriptions (words) | 30 | 1–244 |
| Lengths of dataset descriptions (words) | 105 | 4–512 |
| Lengths of dataset descriptions (sentences) | 8 | 2–34 |

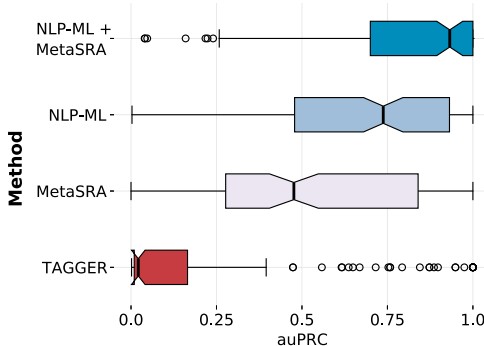

**Fig. 2 | NLP-ML outperforms other text-based methods for sample tissue classification.** Distribution of the area under the precision-recall curve (auPRC) scores across 153 tissues for each of the three individual text-based methods for sample classification: TAGGER, MetaSRA, and NLP-ML. Also shown is the distribution of auPRC scores for combining the predictions of NLP-ML and MetaSRA. Each point in the boxplot is the performance for a single-tissue model averaged across cross-validation folds. In each boxplot (in a different color), the bounds of the box correspond to the distribution's first and third quartiles, the center line is the median, the whiskers extend to the farthest data point within 1.5 times the interquartile range from the bounds, and the separate dots are outliers. Source data are provided as a Source Data file.

each of 13 anatomical systems (Fig. S4). The overall trend is consistent: NLP-ML outperforms MetaSRA in tissues from 10 anatomical systems. The methods perform similarly in nervous system tissues and NLP-ML is worse than MetaSRA for tissues in the respiratory and reproductive systems (Fig. S4). Next, we compared the two methods within four equal-sized groups of tissues where each group of tissues had a similar number of positive training examples. As expected, when more samples are available for training (the last two groups with ≥154 samples per tissue in training), the performance of NLP-ML is high (median auPRC > 0.8) and better than MetaSRA (Figs. S5C, D, S6C, D). In tissues with a moderate number (≥64 and <154) of training samples, NLP-ML and MetaSRA perform quite similarly with a median auPRC of about 0.6 (Figs. S5B, S6B). It is worth noting that MetaSRA achieves an auPRC of close to 1 for a number of tissues in this group. Finally, in the group with the smallest amount of training data, the performance of both methods drop to auPRC <0.5 with NLP-ML performing better than MetaSRA (Figs. S5A, S6A). These patterns indicate that while our method tends to invariably outperform MetaSRA, size of the training set makes a difference. Further, there are likely features in sample text (e.g., names of the fields, i.e., keys) that a rule-based method like MetaSRA may be able to leverage better than our models.

Therefore, we evaluated the performance of combining the predictions from these two methods (see Methods) and observed that combining NLP-ML with MetaSRA results in a median auPRC of 0.93, more than the performance of either method alone (Fig. 2). This overall improvement is particularly due to the boost from MetaSRA in tissues with relatively small training sets (<154 samples per tissue; Fig. S5, S6). The aggregate performance also indicates that automated methods can be developed to accurately assign tissue and cell-type annotations 9 out of 10 times just based on their unstructured or semi-structured text descriptions. A number of specific cases are discussed below (see Discussion and Supplemental Notes) to highlight and understand some ways in which NLP-ML and MetaSRA perform in sample tissue classification.

While the combination of MetaSRA and NLP-ML more accurately annotates samples than NLP-ML, there are a number of factors to consider when using MetaSRA in future applications. MetaSRA is implemented in Python 2, which as of January, 2020, is no longer officially supported. Hence, maintaining a long-term software solution dependent on MetaSRA may be a challenge.

Secondly, the method is low throughput, requiring a large amount of time and computational resources to process a single piece of text. The full MetaSRA pipeline needs to be executed for each unique input. For instance, annotating our >11,000 samples meant generating a unique input for each description and running the full computation pipeline for each input individually. The average runtime for annotating each sample was ~1 h. The average runtime for dataset descriptions often exceeded 3 h. Alternatively, each of our models can create an embedding for any text and make a prediction within seconds with modest hardware requirements. Finally, MetaSRA does not scale with larger passages of text. When making predictions on dataset-level metadata, MetaSRA took hours of computation time. MetaSRA also requires semi-structured data in the form of key-value pairs of information whereas NLP-ML can provide predictions on any piece due to our bag-of-words approach, making NLP-ML more widely usable. We have also shown that, given enough training data, NLP-ML significantly outperforms other text-based methods (see Fig. S5). Therefore, while the combination of NLP-ML and MetaSRA outperforms NLP-ML alone, the scalability and maintainability of NLP-ML make it a more usable and long-term solution (also see Discussion). Additionally, as more data become available for different tissues and cell types, we expect NLP-ML to continue to further improve in performance.

### Incorporating dataset-level information to annotate samples
Current methods for text-based sample annotation are designed to only use the metadata of a particular sample without taking advantage of the metadata available for the parent dataset that the sample belongs to. We investigated if these dataset descriptions could serve as an additional rich source of information to improve the annotation of individual samples. First, for a given dataset, we predicted tissue annotations by providing the dataset's metadata as input to the NLP-ML models trained on sample metadata (excluding samples in that dataset during training; see Methods) and transferred those annotations to each of its constituent samples (referred to as 'dataset-based prediction'). Then, for each sample, we obtained a new tissue prediction by adding the prediction based on its own (sample) metadata with the prediction for its parent dataset (referred to as 'dataset-and-sample-based prediction'). Comparing these schemes to each other, we observe that sample predictions supplemented by dataset predictions result in better performance for a number of tissues and cell types (Fig. 3). This improvement indicates that, for some tissue and cell types, leveraging additional dataset descriptions can help annotate samples correctly. However, the combination of dataset and sample predictions does not always outperform using sample text alone. In fact, the dataset-and-sample-based predictions are nearly always better or are equally as good as using the dataset-based predictions. Therefore, while dataset metadata can be useful in some cases to improve sample annotations, it is insufficient to use dataset metadata alone to make sample tissue annotations. A direct comparison shows that NLP-ML based on sample descriptions alone, in general, outperforms NLP-ML using dataset descriptions alone (Fig. S7, S8).

### Biological meaningfulness and generalizability of NLP-ML models
Having shown that NLP-ML tissue prediction models outperform other text-based methods, we next wanted to explore whether these machine learning models are achieving high accuracy because they are capturing biologically meaningful signals in sample descriptions. First, as each individual tissue model is trained independently of each other, we hypothesized that if the models are capturing meaningful signals in the text data, then models of related tissues should be similar to each other. To test this hypothesis, we analyzed the model coefficients of the full logistic regression classifier of all the tissues using dimensionality reduction (Fig. 4). Visualizing the tissue models based on the

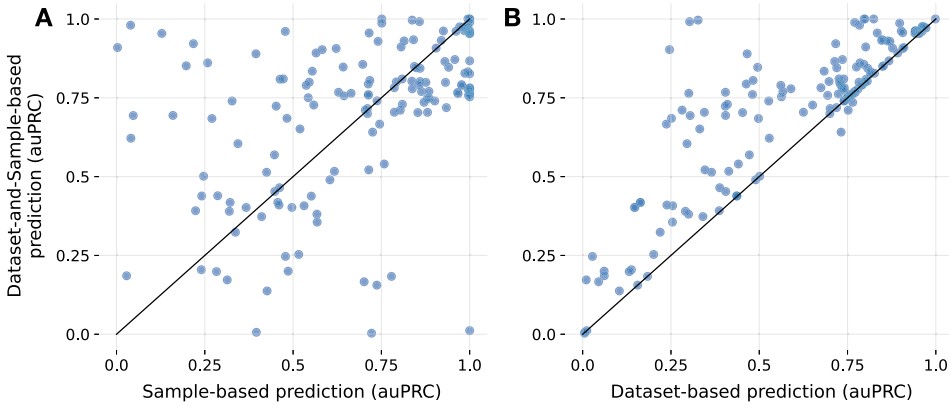

**Fig. 3 | Metadata of sample's parent dataset is useful but insufficient to infer sample annotations. A** Scatterplot of the area under the precision-recall curve (auPRC) scores of sample tissue predictions from just sample text (*x*-axis) vs. combination of predictions from both sample and dataset text (*y*-axis). **B** Scatterplot similar to (and shares *y*-axis with) panel **A** but with auPRC scores of predictions from just dataset text on the *x*-axis. Combination of dataset-and-sample- based predictions versus the dataset-based predictions, where the predicted probability for an experiment is used as the prediction for all samples in that dataset. Each point in the scatterplots correspond to a tissue/cell-type term. auPRC scores are averages across cross-validation folds. The solid line denotes equal performance between the two methods. Source data are provided as a Source Data file.

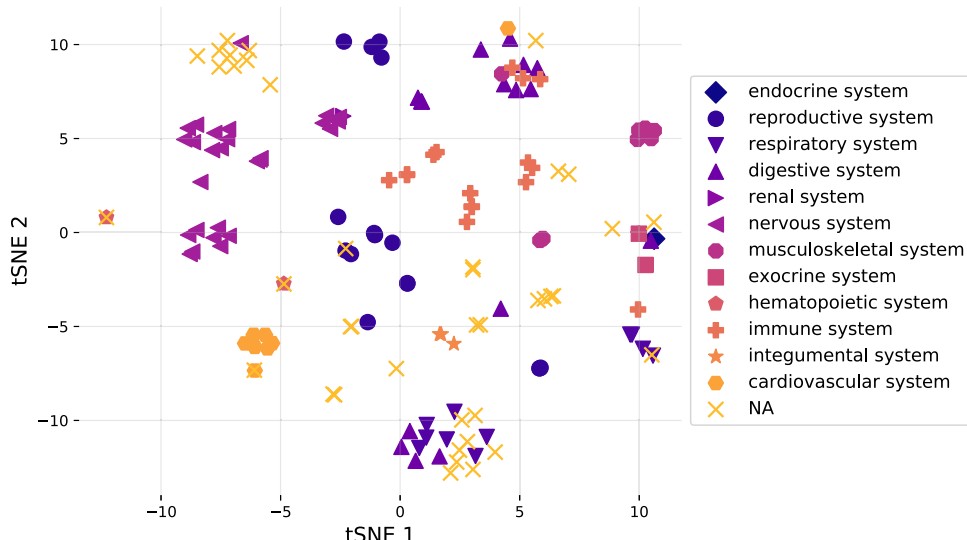

**Fig. 4 | NLP-ML models of anatomically related tissues are similar to each other.** tSNE visualization of standardized logistic regression model coefficients for NLP-ML models of all tissues and cell types trained on the full gold standard. Tissue/cell-type terms with similar models are close to each other on this plot. Colors and marker shapes designate terms to high-level anatomical systems based on the UBERON-CL ontology (see Methods). Source data are provided as a Source Data file.

first two components reveals that models of tissues from the same anatomical systems tend to be clustered with each other, indicating that NLP-ML models of related terms in the tissue ontology have similar model coefficients.

Second, we leveraged the inherent flexibility of our NLP-ML models to examine their generalizability: though they are trained to classify sample descriptions, practically any piece of text can be provided as input to these NLP-ML models to get tissue predictions as output. We began with using descriptions of terms in biomedical ontologies. We created text-based embeddings of terms (see Methods, Creating embeddings for ontology terms) in a number of ontologies based on the names and descriptions of those terms provided in each ontology and made tissue predictions on these embeddings using our NLP-ML models. We then examined the predictions for a subset of tissue-specific Gene Ontology Biological Processes (GOBP) and Disease Ontology (DO) terms based on term-tissue mappings from a previous study[30]. Our NLP-ML models classify tissue-specific GOBP terms to the right tissue with an auPRC three times as expected by random chance (median log2(auPRC/prior) = 3.51 across tissues; Fig. 5A). GOBP terms mapped to blood are the hardest to predict most likely because of the diversity of blood-related processes and the variability in associating specific biological processes to blood. The NLP-ML models have a lower performance (median log2(auPRC/prior) = 1.05 across tissues; Fig. 5B) when classifying diseases (DO terms) to their mapped tissues. This result is not surprising because short descriptions of disease terms available from ontologies will invariably lack information about the tissue associated with the disease. Further, the disease-tissue mappings used for this evaluation are imperfect (e.g., general, multi-system terms such as coenzyme Q10 deficiency being mapped to the brain) or, many times, ambiguous because it is not clear whether a disease is mapped to a tissue because of disease origin, manifestation, or clinical symptoms. In both cases, GOBP and DO, model performance does not depend on the number of terms annotated to a given tissue. When we examine the predicted probability for each tissue-specific term, we find that relevant

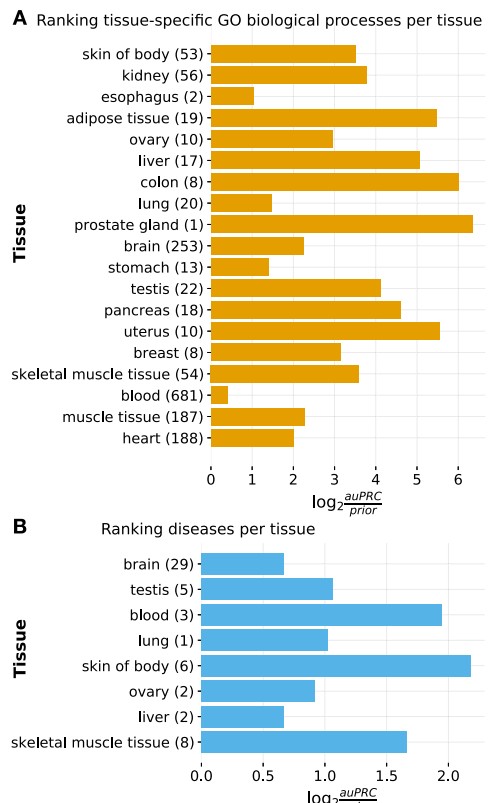

**A** Ranking tissue-specific GO biological processes per tissue

**B** Ranking diseases per tissue

**Fig. 5 | NLP-ML models can correctly classify tissue-associated biological processes and diseases based on their text descriptions. A** Model performances on a set of manually curated tissue-specific Gene Ontology biological process (GOBP) terms[30]. **B** Model performances on a set of manually curated tissue-specific Disease Ontology terms[30]. In both panels, The models are indicated along the *y*-axis with the number of annotated GOBP/DO terms in parentheses next to the tissue name. Performance is shown on the *x*-axis using the logarithm of the area under the precision-recall curve (auPRC) over the prior, where the prior is the fraction of all GOBP/DO terms specific to a particular tissue. This metric accounts for the variable number of annotated terms per tissue. Source data are provided as a Source Data file.

GOBP terms are more often correctly predicted than DO terms (GO: Tables S1–2; DO: TableS3–S4 as examples). Nevertheless, taken together, the model clustering and ontology term predictions demonstrate that the NLP-ML tissue models trained using sample descriptions are capturing meaningful relationships between tissues and are using tissue-relevant signals in text to drive the predictions.

## Comparing and combining NLP-ML with models based on gene expression

A powerful approach to predict sample annotations is to, in fact, completely disregard the provided sample description and instead use the genomics data associated with the sample as input to machine learning models. This approach has been successfully used to predict multiple types of annotations for transcriptome samples including sex[31], tissue and cell types[28], phenotypes[32], and diseases[33]. Therefore, we looked into how our models that predict tissue annotation based on a sample's description (NLP-ML) perform compared to models that predict tissue annotation based on the sample's recorded transcriptome profile. For this analysis, just like before, we trained L1-regularized logistic regression (L1-LR) models—one model per tissue—based on our gold standard of 11,618 microarray samples and their associated tissue labels but using the expression profiles as features (in contrast to using embeddings of sample text descriptions as features in NLP-ML). We observed that L1-LR performed slightly better than

other classifiers for this task (Fig. S9). Then, we compared the performance of these models to the NLP-ML and MetaSRA models using the same CV scheme used to evaluate NLP-ML. With a median auPRC of 0.80, in aggregate, the expression-based models perform a little better than NLP-ML and a lot better than MetaSRA.

Though the performances of expression-based models and NLP-ML models are comparable in terms of overall performances (Fig. 6) (median auPRC values of about 0.80 and 0.74, respectively; *p* = 0.12, Fig. S3), there is high variability in their relative performances over various tissues based on their training set sizes (Figs. S10, S11). Both methods perform with high accuracy for tissues with larger training sets (≥154 samples/tissue; Figs. S10C, D, S11C, D). However, in the group of tissues with a moderate number of training samples (≥64 and <154 samples/tissue), while the performance of both methods drop, the expression-based models achieve high auPRCs for a handful of tissues (Figs. S10B, S11B). This trend is more pronounced in the group with the smallest training sets (<64 samples/tissue; Figs. S10A, S11A). Here, while NLP-ML has a low-to-moderate performance for all tissues in this group, the expression-based models are split into two distinct groups with high and low performance. Hence, at least for a subset of tissues, accurate expression-models can be trained even with very small training sets and, for such tissues, these models can be complementary to the NLP-ML models. Conversely, the better performance of NLP-ML in many other tissues indicates that there are features predictive of tissues/cell types in sample text that cannot be gleaned from the associated expression data until larger training set sizes are reached. Due to this complementarity between these two types of models, we also tested combining their predictions and observed that NLP-ML + Expression models indeed perform better than either model across the board (Figs. S3, S10, S11E–H). Finally, adding predictions from MetaSRA to this combination further improves performance, resulting in a median auPRC of 0.97 (Figs. S3, S10, S11I–L), trends that are also consistent across tissues from various anatomical systems (Figs. S3, S12). In the *Discussion* section below and in Supplemental Note 1, we have considered specific cases in detail to shed more light on when and how these very different kinds of methods—based on text and genomics data—perform differently.

These results reveal that text-based features in sample descriptions can be as effective as and complementary to the molecular signals in their genomics profiles for informing sample tissue prediction. However, text-based models have a unique practical advantage: while models based on genomics profiles need to be trained anew for each experiment type (e.g., microarrays, RNA-seq, ChIP-seq, and methylation arrays) using newly curated gold standard, once trained using any gold standard, text-based models can be used to classify samples irrespective of type.

## Using NLP-ML to annotate samples from multiple experiment types

In our final analysis, we examined two aspects of the predictive performance of our NLP-ML models. First, in addition to the careful dataset-stratified cross-validation scheme, we wanted to evaluate these models using a set of samples that were completely independent of the 11,618 samples used to train any of the models. Second, we wanted to evaluate the ability of our models to crossover and make predictions on samples from new experiment types, a previously mentioned benefit to using text as machine learning features. To satisfy both requirements, we selected the five experiment types on ArrayExpress with the most number of human samples—'RNA-seq of coding RNA', 'ChIP-seq', 'comparative genomic hybridization by array', 'methylation profiling by array', and 'transcription profiling by array'—and made predictions on a large random subset of samples from each experiment type (see Methods, Cross-platform annotations). To then validate our predictions, for each of our five top-performing NLP-ML models—colon, brain, muscle tissue, neural tube, and adipose tissue—

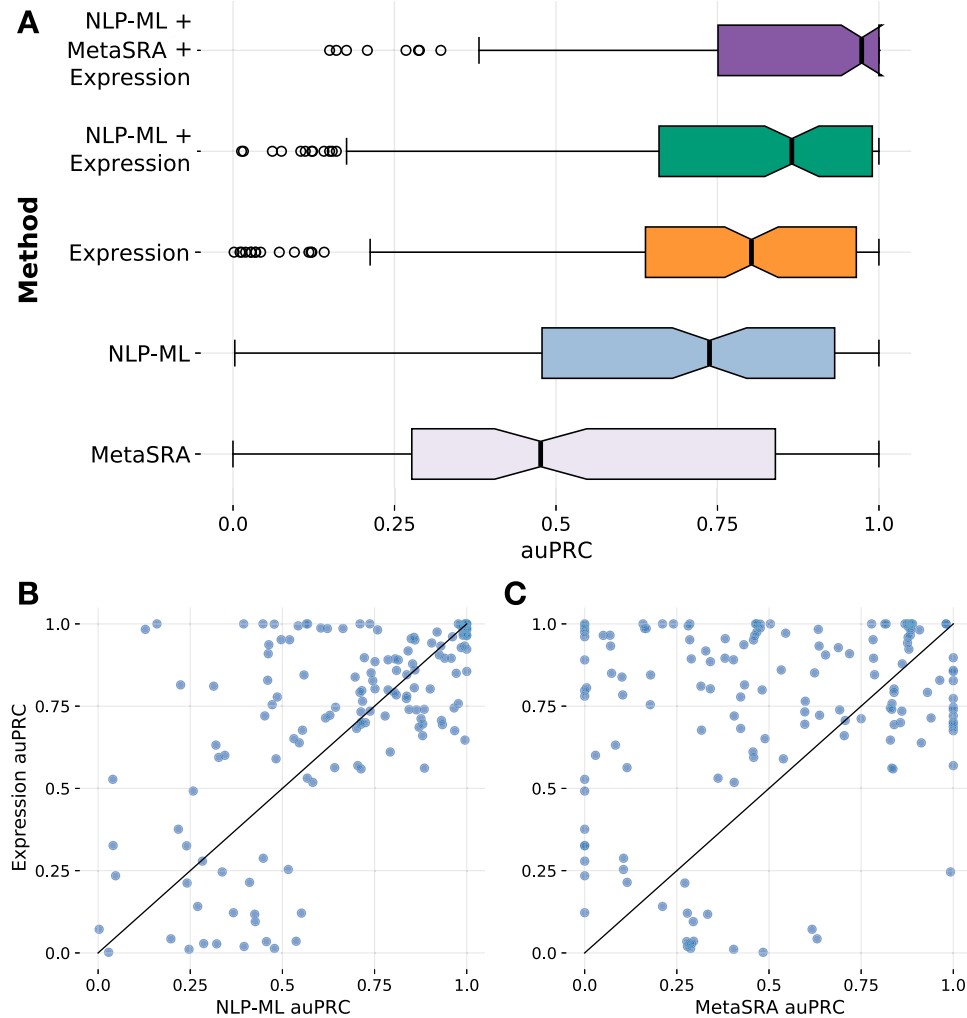

**Fig. 6 | NLP-ML models are nearly as accurate as expression-based models in predicting tissue source of transcriptome samples, and combining them is better than either. A** Distribution of the area under the precision-recall curve (auPRC) scores across 153 tissues for the two top-performing text-based methods—MetaSRA and NLP-ML—and for the method based on expression profiles ('Expression') for sample tissue classification. Also shown are the distributions of auPRC scores for combining the predictions of Expression with NLP-ML ('NLP-ML + Expression') and with NLP-ML and MetaSRA ('NLP-ML + MetaSRA+Expression'). Each point in the boxplot (in a different color; defined as in Fig. 2) is the performance for a single-tissue model averaged across cross-validation folds. **B** Scatterplot of the area under the precision-recall curve (auPRC) scores of sample tissue predictions by NLP-ML models (*x*-axis) vs. predictions by expression-based models (*y*-axis). **C** Scatterplot similar to (and shares *y*-axis with) panel **B** but with auPRC scores of predictions by MetaSRA on the *x*-axis. Each point in the scatterplots correspond to a tissue/cell-type term. auPRC scores are averages across cross-validation folds. The solid line denotes equal performance between the two methods. Source data are provided as a Source Data file.

we manually examined the metadata of the top 10 predictions from each experiment type.

First, we observed that none of the top 10 predictions for any model and experiment type had a predicted probability less than 0.5. Table 2 shows the number of correct predictions out of the top 10 samples made by the five selected models from each of the five experiment types. The average precision (i.e., the average fraction of the top 10 predictions that are correct) of these five models is 0.80, which indicates that NLP-ML models are able to identify tissue samples with high accuracy from new experiment types without needing to retrain the models. Brain and neural tube models performed the best on this cross-experiment analysis, followed by colon and adipose tissue. The low performance of the muscle model (precision = 0.58) could be explained by the presence of the word "musculus" (indicative of a mouse sample) in the majority of samples incorrectly annotated for muscle tissue. This could be the result of a character-level embedding method like flair yielding similar embeddings for similarly spelled words, which may not be an issue for token-level

embedding methods that consider the full word. False-positive predictions by the adipose tissue model were typically samples collected from individuals described to be on high-fat diets. Nevertheless, these shortcomings of NLP-ML are far outweighed by its practical benefit of being able to be deployed seamlessly on samples across experiment types.

## Discussion

Though there is growing awareness about the importance of complete and unambiguous sample metadata[34], the millions of human−omics samples that have already been submitted to databases like NCBI GEO and EBI ArrayExpress are associated with incomplete and unstructured sample descriptions. Here, we propose an approach, NLP-ML, that combines natural language processing and machine learning to annotate samples to their tissue of origin solely based on the unstructured text description available for them.

We have shown that our NLP-ML outperforms two representative text-based methods: (i) TAGGER[29], a baseline method that uses exact-

**Table 2 | NLP-ML models trained on microarray sample descriptions can accurately infer annotations for samples from five different genomics exp types**

|  | RNA-seq | ChIP-seq | Methylation array | CGHA | Microarray | Total |
|---|---|---|---|---|---|---|
| Adipose tissue | 7 | 8 | 10 | 3 | 10 | 38 |
| Brain | 10 | 10 | 10 | 9 | 10 | 49 |
| Colon | 6 | 10 | 10 | 5 | 9 | 40 |
| Neural tube | 10 | 10 | 10 | 7 | 9 | 46 |
| Muscle tissue | 9 | 0 | 10 | 0 | 10 | 29 |

Each row corresponds to one of five top-performing NLP-ML tissue models. The last column shows the total out of 50 that each of these models annotated correctly. Columns 2–6 show the number of samples (out of 10) from each experiment type that each tissue model annotated correctly.

*RNA-seq* RNA-seq of coding RNA, *Methylation array* methylation profiling by array, *CGHA* comparative genomic hybridization by array, *Microarray* transcription profiling by array.

string matching to annotate sample descriptions, and (ii) MetaSRA[24], an advanced method that uses graph-based reasoning to infer annotations from semi-structured sample metadata. We chose a bag-of-words representation for the sample descriptions in this work due to the unstructured nature of most sample metadata in GEO and ArrayExpress. Even when metadata tags or fields are used, they are used and presented inconsistently. Therefore, we chose to use the presence of words in sample descriptions as the input signal.

Examining specific instances illustrates why this approach based on unstructured text (NLP-ML) correctly annotates samples where the other text-based methods (MetaSRA and Tagger) do not. For example, several samples lack informative metadata directly about the source tissues/cell types but the descriptions include mentions of a disease relevant to the source tissue, which is picked up by NLP-ML as a useful piece of information (Supplemental Note 1 1.1, 1.2). NLP-ML also works better than other methods when unstructured sample descriptions are further complicated by details about the experiment and sample preparation (**SN1 2.1, 2.2**). Another common scenario where NLP-ML outperforms other methods is when samples are from individuals with an affliction related to one tissue but the sample itself is from another tissue (typically blood) and both tissues are included in the description (**SN1 2.3**). As expected, in the subset of samples where the tissue is explicitly mentioned without the presence of accessory tissues aside from the true label, all three text-based methods provide accurate annotations not just for microarray samples (**SN1 3**) but also for samples from other technologies (**SN1 3.5, 3.6**). These findings align with the design principles of the other text-based methods we have compared NLP-ML to in this work. Tagger relies on the explicit mention of keywords in the text. MetaSRA has been shown to be highly accurate for providing metadata annotations based on sparse, structured key-value pairs, a result which we are able to observe and confirm in our work as well. Examining specific sample descriptions (Supplemental Note 2) also showcases how NLP-ML is able to achieve lower false-positive rates by taking advantage of the overrepresentation of the true tissue name (compared to mentions of other, non-source tissues or cell types) in the descriptions.

Next, along with the descriptions of the sample alone, we explored the potential of using the typically large, unstructured descriptions that come with the entire experiment a sample is a part of. We observed that experiment-level metadata can bolster NLP-ML predictions for samples from some tissues and cell types (Fig. 3A), indicating that there are experiments where the sample-level information is lacking critical information needed to accurately classify tissues and cell types, and this information could be present in more experiment-wide descriptions (Fig. S8). On the other hand, for most tissues, including the description of the entire experiment did not make any difference or, sometimes, even hurt performance (Fig. 3A). We also show that, in general, experiment-level metadata alone is not sufficient for accurate classification (Fig. 3B, S8) and should be viewed as supplemental rather than complimentary information.

Given the NLP-ML models show good performance in text-based tissue classification, we examined the trained logistic regression models in detail to understand if they were biologically meaningful. Specifically, we compared all tissue models to each other based on their model coefficients and observed that models of tissues from the same anatomical system cluster with each other in low dimensional space (Fig. 4). Then, taking advantage of the fact that our models can be applied to making tissue predictions on any text input, we made predictions on tissue-related pieces texts in the form of descriptions of subsets terms in the Gene Ontology (biological processes) and the Disease Ontology (Fig. 5A, B). Our NLP-ML models achieve high median performance (log2(auprc/prior) values) across all tissues for which we have a set of curated tissue-specific terms[30]. These two findings together show that the NLP-ML models are able to generalize and achieve accurate tissue classification for unseen pieces of text, doing so in a way that captures the underlying biological relationships between tissues.

A number of studies have effectively shown that molecular genomics profile that were experimentally measured from samples, for e.g., the gene-expression profile of a transcriptome sample, can 'predict' various attributes of the sample's source, including tissue, age, sex, and phenotype. This approach is highly complementary to inferring annotation from sample text description because genomics profile-based models can predict attributes missing in the original sample description. Many samples are known to contain descriptions that are lacking informative text or are missing specific attributes altogether[35]. We first confirm this fact in our setting and then show that the aggregate performance NLP-ML is comparable to that of expression-based models in sample tissue classification (Fig. 6A). In most cases, NLP-ML and expression-based model performances across tissues are also correlated with each other except for the tissues with the smallest amount of training data (Fig. S11A–D). Samples where expression-based models tend to outperform text-based methods pertain to multi-level, complex tissues like the brain (**SN1 4.1, 4.2**). Such tissues are also likely to have small training set sizes. Text-based models do not tend to match up to expression-based model performance when sample descriptions contain dense information with several other types of details such as experimental protocols and treatments (**SN1 4.3, 4.4**). On the other hand, for more general tissues involved in complex structures (e.g., hippocampus), or samples with descriptions explicitly containing the tissue terms, as previously noted, our text-based models outperform expression-based models (**SN1 5**).

Given the complementarity of the text- and expression-based models, we combined the predictions from both models and observed that this combination improves the median model performance beyond either NLP-ML or expression alone ($p = 4.47E\text{-}03$ and $p = 0.12$ compared to NLP-ML and expression, respectively). We suspect that the improvement in performance from combining NLP-ML and expression is not greater because both predictors are trained on the

same gold standard. And, hence, if trained on different standards—which can be done given enough available data—we are likely to see even greater performance gains from the combined use of text and molecular-profile features.

Specific instances such as those noted in **SN1 6.1 and 6.6** highlight scenarios in which combining the two models leads to more accurate classification. In these cases, sample metadata contains conflicting tissue signals and the expression profile may have weak molecular signals, but together, result in a correct classification.

Our NLP-ML method has a number of specific advantages compared to existing text and expression-based solutions to annotating samples for tissues and cell types. Our models are able to make predictions for any genomics sample given a plain-text sample description, which lends itself to predictive flexibility compared to methods that use the underlying molecular data to make tissue or cell type annotations. These descriptions can be any unstructured plain text. This is a key advantage over MetaSRA, which was designed for leveraging structured key-value data (particularly the 'Characteristics' field) in order to construct knowledge graphs for annotating samples. NLP-ML is also computationally lightweight: predictions for >300 fully trained models can be made on dozens of pieces of text in a matter of minutes on a modest local computer. This is significantly faster than MetaSRA, which takes on the order of hours for sample descriptions and needs to be executed for each individual piece of text, and Tagger, which needs to load large dictionaries into memory before doing an exhaustive, exact-string matching to the dictionary. MetaSRA was especially designed to operate on very small pieces of text (key-value pairs). Our method outperforms other text-based methods while maintaining biological interpretability both in terms of how the models are trained (taking into account ontology structure when assigning training labels) and in how the models perform (Figs. 4 and 5), which when combined with the other benefits of NLP-ML—predominantly scalability, efficiency, and the ability to work on unstructured text from any source—set it apart from existing text-based methods. Further, because NLP-ML addresses a more general problem, i.e., annotating large collections of unstructured text, it can easily be applied to any text data including descriptions of more –omics data types beyond genomics. However, a significant challenge faced by the biomedical community—an open challenge recognized by funding agencies such as NIH and data consortia such as NCI Data Commons—is the lack of sample-level information that methods like ours can utilize. For instance, PRIDE[36], the preeminent public database for proteomics datasets, only includes descriptions of entire datasets and experiments and not of individual samples. Addressing this challenge of lack of sample-level descriptions requires careful human annotation using semi-automated systems such as ZOOMA (https://www.ebi.ac.uk/spot/zooma/) based on descriptions about samples available elsewhere, including accompanying publications.

The NLP-ML approach developed in this study has several features that lend themselves for further development towards more accurate text-based sample classifiers. First, NLP-ML is accurate for several tissues and cell types despite using all the words in the sample description. However, model performance could be impacted by parts of the metadata that is irrelevant to the prediction task at hand. This could include fields such as institution, platform, and author and contact information. In our current work, we only downweight commonly occuring words using IDF weights computed from PubMed. However, this weighting may end up emphasizing author or institution names that are likely rarely mentioned across PubMed, thus throwing off predictions. In future work, this weighting of words in sample descriptions can be improved in a supervised, task-dependent manner. Second, combining predictions from a sample's molecular profile and the sample's plain-text description can boost performance more than what we have shown here if the two models are trained on

independently labeled samples and datasets. In this way, molecular genomics profile-based models can truly complement text-based methods by filling in information missing in the sample metadata. Third, our observation about the improvement in the performance of NLP-ML with training set size means that prediction accuracy of our approach can continue to be improved simply by curating more samples and datasets to specific tissues and cell types. Finally, the approach presented here can be naturally extended to other types of sample annotations including experimental factors, phenotypes, diseases, and real-valued parameters such as age and condition/treatment duration. Recent studies have also shown similar approaches can benefit query expansion during database searches[37] and in standardizing image metadata[38].

Broadly, our approach empowers new discoveries by providing structured annotations to publicly available genomics samples so that biologists can easily find the samples (and datasets) relevant to their problem of interest from the ocean of hundreds of thousands of samples available to them. Using the annotations from NLP-ML will enable them to more accurately find the samples relevant to their scientific inquiries, which in turn can enable subsequent analyses that may lead to novel discoveries of various forms. Immediate broad impact of NLP-ML can be realized by incorporating it into existing computational sample annotation workflows that bioinformaticians, computational biologists, and data analysts will run on hundreds/thousands of sample descriptions. To ensure that our approach can be easily incorporated into large-scale data workflows, we have released a well-documented Python software (*txt2onto*) and have provided code in our github repository: (i) to train custom text-based machine learning models using NLP-ML (to predict any sample attribute based on the sample description), and (ii) to apply the trained model to predict the desired sample annotations on a large number of new samples. Ultimately, our approach helps democratize data-driven biology by enabling biologists to easily discover publicly available genomics data.

Large consortia of researchers and organizations such as the NIH are rightly pushing for the adherence of FAIR Principles towards making publicly available data findable, accessible, interoperable, and reusable[39]. Our NLP-ML approach supports these goals by providing systematic annotations of human genomics samples for tissues and cell types, thereby enabling researchers to find and reuse relevant data from public data repositories. While care needs to be exercised in using predicted annotations for further analysis[40], such systematic annotations provide a strong starting point towards enhancing the ability to discover, use, and interpret millions of –omics profiles.

## Methods

### Preparation of tissue/cell-type gold standard
**Converting labels from BTO to UBERON-CL.** We obtained tissue and cell type annotations of human gene-expression samples from Unveiling RNA Sample Annotation's (URSA's) >14,000 diverse samples representing over 244 tissues/cell types[28]. In these annotations, the tissues/cell types were identified using terms in Brenda Tissue Ontology (BTO). We mapped these annotations to terms in the UBERON ontology[8] that is also extended to contain terms in the Cell Ontology[41] (CL). We chose UBERON and CL because they are more comprehensive ontologies, agnostic to any specific organism, and updated continuously. The extended UBERON-CL ontology was obtained from https://uberon.github.io/. To map BTO terms to UBERON-CL terms, we first matched as many terms as possible based on exact-string matches. The remaining terms were compared using the difflib Python library to find the closest match in UBERON to the given BTO term using approximate string matching. We then manually reviewed these matches for correctness. In the cases where there were either no quality approximate string matches or no matches at all, BTO terms

were searched on the ontology lookup service[7] (https://www.ebi.ac.uk/ols/index) for UBERON synonyms.

**Assigning positive and negative samples for each tissue.** Next, we used the annotations of samples to UBERON-CL terms and the structure of the UBERON-CL ontology to construct a gold standard for as many tissues and cell types following the procedure in Lee et al.[28]. For a particular tissue or cell-type term, any sample annotated directly to that term of any of its descendants in the ontology is labeled as a positive example. Any sample directly annotated to any of the term's ancestors in the ontology is ignored (removed from the training and testing sets) because of the ambiguity associated with whether or not that sample should be positive or negative. All other samples are labeled as negatives.

We used the sample-level gold standard to then construct a dataset-level gold standard. For each tissue/cell-type term, if the majority of the samples in a dataset were positives for that term, then the entire dataset is labeled as positive. If the majority of samples in the dataset were ignored for that term, the entire dataset is also ignored and thus removed from the training and testing sets. If neither case is true, the dataset is labeled as a negative.

**Assigning tissue and cell-type terms to high-level anatomical systems.** We manually selected a set of high-level terms from the UBERON ontology to categorize all the terms we are building models for into anatomical groups: endocrine system, reproductive system, respiratory system, digestive system, renal system, nervous system, sensory system, gustatory system, hematopoietic system, musculoskeletal system, exocrine system, immune system, integumental system, genitourinary system, cardiovascular system. Then, we mapped each tissue in the ontology to each systems-level term that was its ancestor in the ontology. Terms that do not map to any systems-level terms were assigned to the category of "NA." Once this mapping was complete, we applied a threshold to each system-level term and only kept terms which contained more than 8 models. We determined this cutoff by varying it until 75% of the models were assigned to at least one systems-level term. The remaining models were redesignated as "NA." Thus, the "NA" category comprises models that either do not map to a system-level term, or map to a system-level term that is not prevalent enough in the data. This thresholding is done for simplicity of presenting the results.

**Selecting terms for training and evaluation.** Finally, to ensure robust model training and evaluation, we selected the subset of tissue/cell-type terms that have positively labeled samples from at least three datasets, each with associated metadata from ArrayExpress (see below). As detailed in Model evaluation, during evaluation using cross-validation (CV), samples from the same dataset are kept together in training or testing folds to measure the performance of models on their ability to predict annotations of samples from unseen datasets. This entire curation and selection procedure resulted in a gold standard containing 11,618 human gene-expression samples from the human whole-genome Affymetrix platform from 321 datasets annotated to 153 tissue/cell-type terms (more details in Table 1).

**Downloading and preprocessing sample and dataset metadata**
We extracted sample and dataset metadata from the sample and data relationship files (SDRF) from ArrayExpress as raw text using the curl command line tool. All metadata were downloaded from ArrayExpress on May 29, 2019.

We put together the metadata for each sample by processing the tabularized elements of an SDRF. If a sample appeared in multiple dataset SDRFs, the sample was randomly designated to a single dataset for the entirety of the analysis. A continuous string of metadata text was created for each sample excluding the tabularized headings in an SDRF. This text, with some preprocessing (see below), was used as input for TAGGER and NLP-ML. For MetaSRA, each sample's metadata input JSON file was created by gathering key-value pairs from tabularized elements. Table headers were used for the keys and the respective entries for the sample were used as the values. No additional processing was done with sample JSON files, consistent with the original study.

For dataset-level metadata, we downloaded the investigation description file (IDF) for each dataset in our gold standard and used the entry in the "Experiment Description" field. The IDF and SDRF files were compared to ensure that all samples with metadata also had a corresponding dataset-level metadata. Various attributes of the sample and dataset metadata are presented in Table 1.

**NLP-ML: Natural language processing and machine learning**
The NLP-ML approach described here contains four steps: text preprocessing, creating text-based sample embeddings, training sample tissue classifiers, and predicting tissue annotations for unlabeled samples.

**Text preprocessing.** The entirety of the metadata associated with samples and datasets contains several extraneous pieces of information that need to be preprocessed before being used as input to annotation methods. Hence, we first processed the continuous string of metadata for a sample by removing non-UTF-8 encodable characters and all punctuations. Then, all whitespaces were converted to spaces and the metadata string was split into a list of words on spaces. For each word in the list of words for a sample, if the word contained numerical characters, characters indicative of a URL (e.g., "https," "://", "www"), or was fewer than three characters in length, it was removed from the list. All remaining words were changed to lowercase, lemmatized using the WordNetLemmatizer available in the NLTK Python package[42], and used as the bag of words associated with each sample. For datasets, each sentence in the dataset metadata was processed using the same method.

**Creating text-based sample embeddings.** The second step in NLP-ML entails creating a numerical representation called an embedding for each sample based on its text description by combining the embeddings of the words in the sample's preprocessed metadata. Word embeddings were created using the Flair Python library 0.8.0[27]. We chose Flair because of the availability of multiple neural network models that could be stacked to create word embeddings using combinations of methods. For NLP-ML, we created a stacked model in Flair using ELMo trained on PubMed text[43] and the large uncased version of BERT[44]. Then, we created an embedding for each sample by combining the embeddings of the individual words in the sample's metadata using element-wise weighted averaging.

As potential word weights, we calculated the inverse document frequencies (IDF) for words by processing all PubMed entries (-18 million abstracts and 27 million titles) obtained from https://www.nlm.nih.gov/databases/download/pubmed_medline.html on August 16, 2020. A 'document' was defined as a paper's title and abstract. If an entry lacked one or the other, then a document was considered to be the available text for the entry. For each document in PubMed, we extracted all unique words and incremented their document count by 1. We then calculated the IDF of each word $t$ as

$$\text{IDF}(t,D) = \log \frac{N}{\{d \in D : t \in d\}}, \tag{1}$$

where $N$ is the number of documents in PubMed, $d$ is the number of documents in PubMed—among the total number of documents $D$—that contain the word $t$. We also explored another weighting scheme based on TFIDF (short for term frequency–inverse document frequency),

calculated per tissue/cell-type term by multiplying each word's IDF by the frequency of that word in descriptions of the samples annotated to that term in the gold standard. Finally, the unweighted case is when all weights are set to 1. In all cases, if a word did not have a weight, oftentimes caused by misspelling of words or concatenations of many words (due to metadata entry errors), the word was assigned a weight equal to the mean weight of all words in the description with a weight. Note that an embedding can still be created for these words because Flair creates embeddings using character-level information.

In the case of datasets, for tackling large, multi-sentence descriptions, we created an embedding for each sentence in a dataset's description based on the procedure outlined above, represented the entire dataset as a matrix with the number of rows equal to the number of sentences and the number of columns equal to the dimensionality of the word embedding.

**Training sample tissue classifiers.** The third step in NLP-ML is to use the text-based embeddings as features in a supervised machine learning model built per tissue/cell-type term, trained based on the positively- and negatively-labeled samples in the curated gold standard. After comparisons of multiple classifiers, we chose the L1-regularized logistic regression classifier owing to its good performance, model sparsity, and ability to directly provide prediction probabilities. We trained logistic regression classifiers using the implementation in scikit-learn version 0.20.3 (C = 1, penalty = 'l1', solver = 'liblinear'; Python 3.7.7).

**Predicting tissue annotations for unlabeled samples, datasets, text snippets.** The final part of NLP-ML is to use the trained sample tissue classifiers to classify unlabeled samples based on their metadata. The metadata for these new samples are processed and converted to sample embeddings using the same procedures outlined above (steps 2 and 3). For each tissue/cell-type term, the corresponding trained model provides a prediction score for a particular sample corresponding to a probability (between 0 and 1).

We made predictions for datasets based on dataset-level metadata by first splitting the full block of dataset text into individual sentences. The sentences are then processed, embedded, and supplied to trained NLP-ML models for predictions. For each tissue/cell-type term, the prediction score for a particular dataset corresponds to the maximum predicted probability across all sentences.

We obtained the dataset-and-sample-based predictions for each sample by simply adding the predicted probabilities based on its own (sample) metadata with the probability for its parent dataset. We compared this addition with taking the maximum of the two (see Figs. S4, S5) and observed no significant difference.

To make predictions on text that was not sample or dataset metadata, e.g., descriptions of biological processes or diseases,

## Other text-based methods for inferring sample tissue annotation

**TAGGER.** We downloaded the full TAGGER dictionary from http://download.jensenlab.org/. We added additional terms to this dictionary based on UBERON and CL terms in the extended Uberon Ontology, including all synonyms present in each term's metadata. Using the Python 2 module for TAGGER, we identified UBERON and CL matches in the sample metadata available from ArrayExpress. For each tissue/cell-type term, we set the TAGGER prediction score for a particular sample as 1 if that term appears in the list of terms identified by TAGGER. If not, the sample's score is set to 0.

**MetaSRA.** We downloaded the source code for the MetaSRA pipeline from https://github.com/deweylab/MetaSRA-pipeline on May 16, 2019. A modification to the source code was made to save the many constructed graph objects as a pickle file to reduce computation time. The

OBO files used in MetaSRA were downloaded between May 16, 2019 and May 31, 2019. MetaSRA takes unprocessed sample metadata in the form of key-value pairs as input and outputs a list of ontology terms. For each tissue/cell-type term, we set the MetaSRA prediction score for a particular sample as 1 if that term appears in the list of terms identified by MetaSRA. If not, the sample's score is set to 0.

## Expression-based sample tissue classification

We used GEOmetadb[45] to get the list of available human Affymetrix whole-genome gene-expression samples in GEO. We queried and downloaded all CEL files corresponding to these samples on July 29, 2019. These data were background corrected, normalized, and summarized using the frma[46] and affy[47] R packages and gene names converted to Entrez IDs using the hgu133afrmavecs, hgu133plus2frmavecs, hgu133plus2hsentrezgcdf, and hgu133ahsentrezgcdf R packages[48]. Then, we trained expression-based sample tissue classifiers using L1-regularized logistic regression (identical to the models used in NLP-ML) with these expression profiles as features and based on labeled samples in the curated gold standard.

## Combining predictions from different methods

When combining predictions between any set of methods among NLP-ML, MetaSRA, and Expression, we first calculated an F1 score (the harmonic mean of precision and recall) for each method for a particular tissue or cell type. These F1 scores were then used as weights to calculate a weighted average of the predicted probabilities for a particular sample.

## Model evaluation

We evaluated all methods using dataset-level $k$-fold cross-validation (CV). Dataset-level CV means that all the samples within the same dataset are never split across folds, i.e., kept together in the training or the testing folds. We do this to prevent the models from learning any dataset-specific characteristics and evaluate the ability of the models to predict annotations of samples in unseen datasets. As our gold standard contains several tissues/cell-type terms with as low as three positively labeled datasets, we picked 3–5 CV folds for each term's model based on data availability: 3-fold CV for terms with just three positive datasets, 4-fold CV for terms with four datasets, and 5-fold CV for terms with five or more datasets. Results reported in this work are based on model performance calculated using area under the precision-recall curve (auPRC) and averaged over the 3-, 4-, or 5-folds depending on the term.

For all evaluations, predictions on samples and datasets were made by models trained on folds that did not include any sample from the corresponding datasets. Predictions on descriptions of ontology terms and metadata of samples from different technologies were made by models trained on the full gold standard.

## Creating embeddings for ontology terms

To test the ability of the trained NLP-ML models to generally classify text to various tissues, we supplied the descriptions of biological processes and diseases (associated with various tissues) as input to NLP-ML. We obtained these descriptions by parsing the OBO files for the Gene Ontology (Biological Process branch; GOBP) and the Disease Ontology (DOID) and extracting the plain-text name and definition for each ontology term. For each ontology term, we concatenated the name and definition to a single string, and then preprocessed and embedded each string following the same methodology outlined for sample descriptions to generate ontology term embeddings.

## Cross-platform annotations

To explore the ability of NLP-ML models (trained on descriptions of gene-expression samples) to annotate genomics samples from different technologies, we chose five of our top-performing NLP-ML models

from our cross-validation results: colon (UBERON:0001155), brain (UBERON:0000955), muscle tissue (UBERON:0002385), neural tube (UBERON:0001049), and adipose tissue (UBERON:0001013). Then, we chose five experiment types in ArrayExpress with the maximum number of available human samples: RNA-seq of coding RNA, ChIP-seq, comparative genomic hybridization by array, methylation profiling by array, and transcription profiling by array. We downloaded and preprocessed all available sample metadata from all SDRFs across the samples from these experiment types and randomly selected 10,000 samples from each experiment type. We constructed a sample embedding for all 50,000 samples and ran them through the five fully trained NLP-ML models (top five in performance) to make a prediction for each of these five tissues/cell types. Then, for each of our five models, we selected the top 10 predicted samples from each experiment type and pulled the corresponding processed metadata from our input files. We then manually examined the metadata from these 50 samples in a random order for each tissue/cell-type model to evaluate the models' predictions. Correct and incorrect predictions were declared as true positives and false positives. Additionally, in the case where a sample's metadata was too ambiguous to decide whether the annotation was correct or not, or if the metadata lacked any usable information, these cases were also declared as false positives.

### Reporting summary

Further information on research design is available in the Nature Research Reporting Summary linked to this article.

## Data availability

All the gold-standard annotations (labeled examples) used to train all the models and the cross-validation splits used to evaluate the models are available in the repository https://github.com/krishnanlab/txt2onto. Lists of samples from specific microarray platforms (available at https://github.com/krishnanlab/txt2onto/tree/main/gold_standard) were downloaded from GEOMetaDB https://doi.org/doi:10.18129/B9.bioc.GEOmetadb. Metadata were downloaded from ArrayExpress https://www.ebi.ac.uk/arrayexpress/. Source data files necessary to recreate our figures are provided in the repository and with this paper. Source data are provided with this paper.

## Code availability

We have made the trained NLP-ML models, a Python utility for text-based tissue classification, and demo scripts, along with extensive documentation, at https://github.com/krishnanlab/txt2onto (v1.0.0 archived at https://doi.org/10.5281/zenodo.7232237)[49]. Given an input file where each line is a piece of text to be classified, the *txt2onto* utility will perform the necessary text preprocessing, create an embedding for each piece of text, and then run each embedding through our pretrained tissue models. The repository also includes a set of utilities for training new NLP-ML models for a user-defined problem.

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

## Acknowledgements

We would like to thank Kayla Johnson for their mentoring, support, and feedback on the manuscript, and all members of the Krishnan Lab for valuable discussions and feedback on the project. This work was primarily supported by US National Institutes of Health (NIH) grants R35 GM128765 to A.K. and in part by MSU start-up funds to A.K. and MSU Rasmussen Doctoral Recruitment Award and Engineering Distinguished Fellowship to N.T.H.

## Author contributions

N.T.H., M.M., and A.K. designed the study. N.T.H., M.M., and A.Y. developed the software. N.T.H., M.M., L.A.G., and A.K. performed the analyses. N.T.H. and A.K. interpreted the results. N.T.H. and A.K. wrote the final manuscript with feedback from A.Y.

## Competing interests

The authors declare no competing interests.
