## [Peer Review File · Nature Communications]

Systematic tissue annotations of genomics samples by modeling unstructured metadataREVIEWER COMMENTS

Reviewer #1 (Remarks to the Author):

In this manuscript, Hawkins et al. proposed a new approach to standardizing annotations of -omics samples through an analysis of their textual metadata. The core methodology described in this work is well supported by the community; there is an adaptable foundation for metadata standardization. Incorporation of embeddings is a technical advancement on this front but the exact benefits conferred by this approach remain opaque. The primary advantages of this approach remain to be defined (e.g., as compared to MetaSRA, its improved speed and accuracy?) Is NLP-ML more flexible in terms of its ready applicability to more -omics data types beyond gene expression? Some further benefits are noted in the Results regarding NLP-ML vs. MetaSRA; the potential impact of these improvements warrants further analyses and investigations.

Issues/Comments:

1) How does this new NLP-ML method address the high false positive issue of NER-based methods (without manual curation step) when one dataset contains multiple tissue/cell-type experimental data?

2) In the Abstract as well as in the Introduction, the authors refer to -omics data as a whole but only the study itself only concerns two resources and primarily on genomic/transcriptomic datasets. Please replace the wording of “-omics” throughout the manuscript, as this project focused on 1 or 2 types of data sources, and it is not applicable to a variety of -omics data types. Additionally, there is a lack of evidence/data to demonstrate NLP-ML’s capacity to tackle other sorts of -omics data. Each type of -omics data presents its own inherent technical challenges; we do not see that they are necessarily commonly comparable to genomic or transcriptomic data.

3) An online demonstration, beyond that provided by the example scripts on the project’s GitHub repository, would be very useful for understanding potential use cases for NLP-ML.

4) The Zotero reference import process appears to have created some reference issues, e.g., the references for ZOOMA (as internet resource) and ConceptMapper (as conference proceedings) are missing dates and other info. Some hyperlinks embedded in references lead to incorrect locations.

5) Knowing the time factor necessary for down-loading PubMed corpus, as described in “Creating text-based sample embeddings”, is critical to reproducibility. It is important for the authors to consider high/low frequency; were they excluded when building embeddings?

6) Expression based tissue classifications use distinct features; alternative methods, other than L1-regularized logistic regression, may support a fair comparison.

7) Is averaging auPRC weighted? The sample size of the 3-5 CV is different.

8) Page 2, paragraph 2. “While they often provide high-quality annotations, automated high-throughput methods are needed to ...”: This transition here is a bit confusing. Here, “they” refers to the “manual curation”, not a more grammatically natural “automated high-throughput methods”. Some rephrasing might improve the clarity and flow.

9) Page 3, paragraph 1. “NER-based methods suffer from several false positives”: Is “several” a typo for “severe” or meant “a certain level” of false positives?

10) As described in the methods: “We obtained tissue and cell-type annotations of human gene-expression samples from Lee et. al., 2013 (Lee et al. 2013).” Are those samples including situations where varied and conflicting pieces of information exist in sample descriptions (e.g., mentioning more than one tissue/cell-type)? Are these samples from Lee’s publication limited to blood cells and lymphoid tissue-derived cells (Suppl. Fig 2 in Lee’s publication), or the URSA’s >14,000 diverse

samples representing over 244 tissue/cell-type terms? Please clarify which one was used for training.

Reviewer #2 (Remarks to the Author):

There are millions of publicly available -omics samples, but much of their metadata consists of unstructured free-text fields. Converting unstructured biological metadata into structured metadata has been done in the past largely via text-matching and graph-based annotation. Some authors have used neural networks to attempt to label samples, but so far such methods have struggled to predict tissue and cell type information.

In this work, Hawkins et al. develop a model (txt2onto) by fine-tuning a transformer on text from PubMed, embedding sample and dataset-level metadata using the transformer, and using l1-regularized logistic regression to predict terms in the UBERON-CL ontology for individual samples.

They then evaluate txt2onto in a number of ways. They compare their model's performance against competing tools, MetaSRA and Tagger, finding that their model outperforms these methods. The authors also find that txt2onto predicts tissues of expression samples almost as well as models trained directly on the expression data. Further, they validate that the model is learning something biologically meaningful by showing that it maps anatomically related tissues close together in t-SNE space, and that the model generalizes well enough to predict tissue-specific biological processes.

While the paper is convincing, a few comments remain:

Minor comments:

1) While the authors have published the code required to run the pretrained version of txt2onto, they do not make the code required to train the models available.

2) It is unclear why the word 'musculus' would cause false positives for muscle as mentioned in the "Using NLP-ML to annotate samples from multiple experiment types" section. The original ELMo paper shows that ELMo embeddings can disambiguate between different definitions of the word "play" based on context, so I wouldn't expect words to have similar embeddings caused by similar spelling. Is it possible that the mouse samples in the training set are frequently annotated as being muscle samples?

3) I appreciate the authors' use of dataset-aware train/validation set splitting to avoid data leakage and overoptimistic estimates of performance

4) Table 2 may not be a representative sample of the models' predictions? Most of the models do well for the predictions they're most confident in, but would the results for, e.g., ten random predictions with predicted probability > .9 show the same distribution? This experiment may be infeasible to run due to the manual evaluations required though.

5) It is unclear why the authors decided on a one-vs-rest formulation for the logistic regression model instead of using multi-class logistic regression.

Reviewer #3 (Remarks to the Author):

Hawkins et al present a method "NLP-ML" to infer tissue and cell type annotations from free text metadata, and compare the performance of their system to two well established approaches, TAGGER and MetaSRA, as well as direct tissue annotation from expression data. Demonstrated performance is better than the other two text-based methods, but slightly less good than the expression database based approach. The authors argue that NLP-ML nevertheless improves on the

state of the art, as the method

a) helps to improve performance overall in combination of multiple approaches, and

b) is relatively easy to apply, and might be useful for the annotation of multiple omics data types, as it relies only on free text metadata, rather than highly structured expression data.

The method is well described and documented, including source code and available/referenced datasets.

* Major concerns:

The authors claim, even in the title, a potential to generalise the method to "omics" samples, which is the major claim to progress beyond state of the art, compared to expression based methods.

However, this is only demonstrated based on data from two databases (Geo, ArrayExpress) and five relatively "related" methods. To support this claim of potential to generalise, it would be helpful to apply the method to data from a different database and field. Proteomics would be a potential example, as there are enough public datasets available, and the sample character is still related. Metabolomics would be a more challenging demonstrator, both in terms of data availability and divergence of sample types.

Minor concerns:

Somewhere early on, "available samples" should be defined, it might be interpreted as physical samples available from providers. The subject of this manuscript are "available sample descriptions". On a side note, not a mandatory revision, a really interesting extension of the manuscript might be to map the samples to actually available samples from biobanks etc.

P2: "continues to grow exponentially". Do available samples really grow exponentially?

P2: ArrayExpress ref might be updated.

On pages 3 and 14, the authors mention that MetaSRA is "slow", "low throughput" in comparison to NLP-ML. This should be supported by objective measures.

P3, line 3: "...several false positives". Several is an odd quantification here.

P4: Add literature reference for Ontology Lookup Service.

P18: "perform similarly overall" is a bit of an idealising statement.

Fig 6, panel A, top row; fig S10, panel B, top row: The boxplots show a strange artifact at the right border, exceeding "1". While probably a problem of the underlying library, it would be nice to correct this for a potential next version.

REVIEWER COMMENTS

Reviewer #1 (Remarks to the Author):

In this manuscript, Hawkins et al. proposed a new approach to standardizing annotations of -omics samples through an analysis of their textual metadata. The core methodology described in this work is well supported by the community; there is an adaptable foundation for metadata standardization.

Comment 1.1:

Incorporation of embeddings is a technical advancement on this front but the exact benefits conferred by this approach remain opaque. The primary advantages of this approach remain to be defined (e.g., as compared to MetaSRA, its improved speed and accuracy?) Is NLP-ML more flexible in terms of its ready applicability to more -omics data types beyond gene expression? Some further benefits are noted in the Results regarding NLP-ML vs. MetaSRA; the potential impact of these improvements warrants further analyses and investigations.

Response: We agree with the reviewer that the advantages of NLP-ML can be discussed in more detail. Hence, we have now included a new paragraph in the *Discussion* section:

“Our NLP-ML method has a number of specific advantages compared to existing text and expression-based solutions to annotating samples for tissues and cell types. Our models are able to make predictions for any genomics sample given a plain-text sample description, which lends itself to predictive flexibility compared to methods that use the underlying molecular data to make tissue or cell type annotations. These descriptions can be any unstructured plain text. This is a key advantage over MetaSRA, which was designed for leveraging structured key-value data (particularly the ‘Characteristics’ field) in order to construct knowledge graphs for annotating samples. NLP-ML is also computationally lightweight: predictions for >300 fully trained models can be made on dozens of pieces of text in a matter of minutes on a modest local computer. This is significantly faster than MetaSRA, which takes on the order of hours for sample descriptions and needs to be executed for each individual piece of text, and Tagger, which needs to load large dictionaries into memory before doing an exhaustive, exact-string matching to the dictionary. MetaSRA was especially designed to operate on very small pieces of text (key-value pairs). Our method outperforms other text-based methods while maintaining biological interpretability both in terms of how the models are trained (taking into account ontology structure when assigning training labels) and in how the models perform (Figure 4 and 5), which when combined with the other benefits of NLP-ML – predominantly scalability, efficiency, and the ability to work on unstructured text from any source – set it apart from existing text-based methods. Further, because NLP-ML addresses a more general problem, *i.e.*, annotating large collections of unstructured text, it can easily be applied to any text data including descriptions of more –omics data types beyond gene expression.”

Issues/Comments:

Comment 1.2:

How does this new NLP-ML method address the high false positive issue of NER-based methods (without manual curation step) when one dataset contains multiple tissue/cell-type experimental data?

Response: This is a good question. We, in general, control for false positives by optimizing our models for *area under the precision recall curve* (auPRC) when selecting our machine learning models (Figure S2). By doing so, we are optimizing for a high precision, which in turn directly reduces our false positive rate. In terms of how the natural language processing part minimizes false positives, we have a couple of working hypotheses. We examined all of the cases where for a given sample and for a particular tissue or cell type, the true label according to our gold standard is negative, and our method (NLP-ML) correctly labels the sample as such (true negative) but either MetaSRA or Tagger labels the sample as a positive (false positive). We then filtered these instances to ones where the predicted probability from NLP-ML is < 0.05 to examine cases where our models were confident in assigning a negative label, and further filtered these cases to instances pertaining to a tissue or cell types whose auPRC from cross validation is > 0.80 to only consider tissues and cell types where the predicted probabilities from NLP-ML are most likely to be accurate. Below, we describe our observations from three specific tissues or cell types along with a count of the number of samples that fulfilled the above criteria.

For brain (N = 12), for all of the cases where NLP-ML correctly identified a non-brain sample correctly as a negative but the other text-based methods did not, the samples in question came from liver or blood, but all came from either patients who are brain dead or patients with brain cancer. For liver (N = 26), the true label for the samples were either blood or colon (specifically samples from colon adenocarcinoma tumors), but the patients were either liver transplant patients in the case of the true label being blood, or the word “liver” just appears in the sample description. For intestine (N = 23), all samples were from stomach stromal tumor, but terms like “gastrointestinal” and “small intestine” are mentioned throughout.

These instances point to one hypothesis about how NLP-ML might have been able to correctly label these samples as negatives for the appropriate tissues. In almost all cases where NLP-ML correctly predicts a negative and the other text-based methods incorrectly assign a positive label, the true label tissue name appears more times than any other tissues or cell types in the description. This hypothesis is supported by how we generate a text-based feature vector for a sample based on its ‘bag-of-words’ (from the description) where more frequently appearing words directly contribute more strongly to the final feature vector, making it more associated with the correct tissue name and less associated with the incorrect ones. We also suspect that there may be some words present in the description’s bag of words that provide additional contextual clues that can additionally point NLP-ML to the true tissue of origin, thus contributing to NLP-ML’s lower false positive rate.

Having made these observations based on careful manual inspection, because the predictions are made based on dense vectors that combine signals from all the words in a description, explaining individual prediction results of NLP-based models remains an open area of investigation in our work (and in others’ research as well) and we will pursue it in the future.

Comment 1.3:

In the Abstract as well as in the Introduction, the authors refer to -omics data as a whole but only the study itself only concerns two resources and primarily on genomic/transcriptomic datasets. Please replace the wording of “-omics” throughout the manuscript, as this project focused on 1 or 2 types of data sources, and it is not applicable to a variety of -omics data types. Additionally, there is a lack of evidence/data to demonstrate NLP-ML’s capacity to tackle other sorts of -omics data. Each type of -omics data presents its own inherent technical challenges; we do not see that they are necessarily commonly comparable to genomic or transcriptomic data.

Response: This is a fair point that is also made by Reviewer 3. In principle, our method is general and can be applied to annotate samples from any –omics experiment type beyond genomics/transcriptomics. The reviewer is right that each –omics molecular data presents its own technical challenges. However, as our approach annotates each sample only based on its text description and not based on the recorded molecular data (e.g., expression profile or methylation profile), these technical variations will not affect NLP-ML. If there are systematically different ways in which researchers *describe* samples from different –omics types, that might affect our method, but currently there is no evidence for such systematic textual differences.

Nevertheless, as we have detailed below (*‘Validation on other omics types’*), without a gold standard to evaluate our model predictions against and the infeasible manual curation effort required to validate predictions based on external information (e.g., information hidden away in the papers describing the datasets), it is not possible to unequivocally claim that our method works well across all –omics types.

Therefore, we have revised the wording in our manuscript to now only make claims pertaining to predicting tissues and cell types for *genomics* data rather than *-omics* data in general. The term “-omics” remains in the manuscript in a few areas where the discussion is broader than just the results conveyed from our work, but claims pertaining to our NLP-ML models have been revised.

Validation on other omics types: In an effort to validate our NLP-ML predictions on samples from other omics data types, we examined proteomics data from two databases: PrideDB (<https://www.ebi.ac.uk/pride/>) and OmicsDI (<https://www.omicsdi.org/>), which contain metadata for a large number of proteomics and -omics experiments, respectively. PrideDB not only contains plain text metadata for each proteomics experiment, but in many cases, the experiments also include tissue annotations. These annotations, presumably, are submitted by experimenters with their submission to the database. However, PrideDB does not include any sample-level information. Instead, the database contains sample protocol information, which is more akin to a paragraph from the Methods section of an accompanying manuscript and less indicative of actual sample-level information. We sought to use OmicsDI for doing a broader -omics evaluation, but the proteomics data from OmicsDI is pulled directly from PrideDB, and many larger -omics types lack gold standard labels like we would have working with PrideDB metadata. Evaluating our approach on experiment protocol description – which are more similar to text in Methods sections describing the full experiment and less to descriptive sample metadata – will be unfair. Conducting a fair evaluation entails a substantial manual curation

process. Therefore, we have revised the wording of our manuscript to only make claims about genomics samples.

Comment 1.4:

An online demonstration, beyond that provided by the example scripts on the project's GitHub repository, would be very useful for understanding potential use cases for NLP-ML.

Response: This is a good idea. We have included a number of different tools in the repository as per the reviewer's suggestion.

1. Training custom NLP-ML models: Users can now train their own NLP-ML models using labeled text of their choosing using our method (<https://github.com.krishnanlab.txt2onto/blob/main/README.md#use-case-2-training-new-nlp-ml-models>). This means that users can either train new models by subsetting from our gold standard for a particular tissue or cell type, or they can create their own curated dataset for their own text classification problem of interest and use our code to train brand-new NLP-ML models.
2. Running predictions using custom or pre-trained models on any text: Users can make predictions on any new piece of text using either their own trained models or our pre-trained models (<https://github.com.krishnanlab.txt2onto/blob/main/README.md#use-case-1-making-predictions-on-unstructured-text-using-nlp-ml>).

The repository README has been updated to reflect these changes and new scripts have been included.

Comment 1.5:

The Zotero reference import process appears to have created some reference issues, e.g., the references for ZOOMA (as internet resource) and ConceptMapper (as conference proceedings) are missing dates and other info. Some hyperlinks embedded in references lead to incorrect locations.

Response: This has been fixed. We have double checked and corrected all hyperlinks in the bibliography.

Comment 1.6:

Knowing the time factor necessary for down-loading PubMed corpus, as described in "Creating text-based sample embeddings", is critical to reproducibility. It is important for the authors to consider high/low frequency; were they excluded when building embeddings?

Response: These are important points. We have now included in the *Methods* section the dates when our PubMed corpus was downloaded (16 August 2020) and when the corpus used to train

the ELMo models (used by Flair) were potentially downloaded (to the best of our knowledge, based on the ELMo publication: <https://aclanthology.org/N18-1202/>) (2018).

Comment 1.7:

Expression based tissue classifications use distinct features; alternative methods, other than L1-regularized logistic regression, may support a fair comparison.

Response: This is certainly true. Therefore, we already evaluated 4 different machine learning algorithms for expression-based tissue classification: XGBoost, Random Forest, Support Vector Machine, and both L1- and L2-regularized Logistic Regression. These results were presented in our Supplemental Figure 9. For XGBoost, Random Forest, and Support Vector Machine, the default parameters from scikit learn were used. We compared the performance of these 5 classifiers and examined the model performances without considering training size (Figure S9. A) and considering training size (Figure S9. B-E). In all cases, the classifier performances were nearly identical with some slight variation depending on sample size. Based on this comparison, we chose to use L1-regularized Logistic Regression for expression-based classification, same as the choice we made for text-based classification (Figure S2).

Comment 1.8:

Is averaging auPRC weighted? The sample size of the 3-5 CV is different.

Response: For each tissue and cell type, we split the available data in our gold standard into a variable number of folds depending on the number of experiments with positively annotated samples in it. To ensure our models do not learn any dataset-specific signals, we never split samples from the same experiment across different folds. Therefore, to train a model for the purposes of evaluating NLP-ML performance for annotating for a particular tissue or cell type, we needed that tissue or cell type to have a minimum number of experiments with positively annotated samples in it. In general, a larger number of folds in CV lends itself to better generalizability on unseen data because overfitting to the training data is averaged out between the different folds. However, to train models for as many tissues as possible, we set the minimum number of experiments with positively annotated samples to 3. Since some tissues and cell types have a large amount of positively labeled training data, we also wanted to set a maximum number of folds in CV for computational purposes alone. Therefore, we ended up choosing to do a variable number of folds ranging from 3 to 5 for each tissue. When reporting the final results, we average the auPRC values across the k folds for each tissue or cell type to show the average performance (where k is 3, 4, or 5). The auPRC values from different k 's are not weighted when calculating an aggregate model performance. Instead, we first split the terms into four bins where terms within each bin had similar training set sizes (and, hence, identical or similar k 's) and then assessed the relative performance of methods within these bins. Examples include Figures S2, S5, S6, and S7. Also, when assessing the overall performance of methods, we performed paired tests that compare methods on a term-by-term basis (each with a particular k) when estimating statistical significance (Figure S3).

Comment 1.9:

Page 2, paragraph 2. “While they often provide high-quality annotations, automated high-throughput methods are needed to ...”: This transition here is a bit confusing. Here, “they” refers to the “manual curation”, not a more grammatically natural “automated high-throughput methods”. Some rephrasing might improve the clarity and flow.

Response: We have revised the language to the following: “While curation efforts often lead to high-quality gold standards, automated high-throughput methods are needed to keep up with the scale of available samples, which already totals to more than a million and continues to grow exponentially (Krassowski et al. 2020; Conesa and Beck 2019; Perez-Riverol et al. 2019; Stephens et al. 2015).”

Comment 1.10:

Page 3, paragraph 1. “NER-based methods suffer from several false positives”: Is “several” a typo for “severe” or meant “a certain level” of false positives?

Response: We have revised the sentence to the following: “However, without an additional step of manual curation, NER-based methods suffer from high false-positive rates due to the presence of varied and conflicting pieces of information in sample descriptions.”

Comment 1.11:

As described in the methods: “We obtained tissue and cell-type annotations of human gene-expression samples from Lee et. al., 2013 (Lee et al. 2013).” Are those samples including situations where varied and conflicting pieces of information exist in sample descriptions (e.g., mentioning more than one tissue/cell-type)?

Response: As noted in our response to Comment 1.2, the sample descriptions included here did indeed contain varied and conflicting pieces of information, including mentions of more than one tissue/cell-type terms within a given description.

Comment 1.12:

Are these samples from Lee’s publication limited to blood cells and lymphoid tissue-derived cells (Suppl. Fig 2 in Lee’s publication), or the URSA’s >14,000 diverse samples representing over 244 tissue/cell-type terms? Please clarify which one was used for training.

Response: The language has been changed to clarify this point: “We obtained tissue and cell type annotations of human gene-expression samples from Unveiling RNA Sample Annotation’s (URSA’s) >14,000 diverse samples representing over 244 tissues/cell types (Lee et al. 2013).”

Reviewer #2 (Remarks to the Author):

There are millions of publicly available -omics samples, but much of their metadata consists of unstructured free-text fields. Converting unstructured biological metadata into structured metadata has been done in the past largely via text-matching and graph-based annotation. Some authors have used neural networks to attempt to label samples, but so far such methods have struggled to predict tissue and cell type information.

In this work, Hawkins et al. develop a model (txt2onto) by fine-tuning a transformer on text from PubMed, embedding sample and dataset-level metadata using the transformer, and using L1-regularized logistic regression to predict terms in the UBERON-CL ontology for individual samples.

They then evaluate txt2onto in a number of ways. They compare their model's performance against competing tools, MetaSRA and Tagger, finding that their model outperforms these methods. The authors also find that txt2onto predicts tissues of expression samples almost as well as models trained directly on the expression data. Further, they validate that the model is learning something biologically meaningful by showing that it maps anatomically related tissues close together in t-SNE space, and that the model generalizes well enough to predict tissue-specific biological processes.

While the paper is convincing, a few comments remain:

Minor comments:

Comment 2.1:

While the authors have published the code required to run the pretrained version of txt2onto, they do not make the code required to train the models available.

Response: We agree that this would be useful. We have now included code to train custom NL-ML models in our github repository (<https://github.com/krishnanlab/txt2onto/blob/main/README.md#use-case-2-training-new-nlp-models>). We have included two files for this particular application. The first set of tools will enable users to create a training set input in the correct format for any tissue or cell type in our gold standard. This will allow users to retrain all the full models included as a part of our work. Alternatively, users can provide their own correctly-formatted, labeled text to train models for their own purposes using our methodology. Such models could be for any text classification task with sufficient labeled data with the caveat that our framework has been tested only within the scope of annotating biomedical text for tissues and cell types. When making predictions on new text, users can now specify whether to use the available fully-trained models from our work or their own, custom models.

Comment 2.2:

It is unclear why the word 'musculus' would cause false positives for muscle as mentioned in the "Using NLP-ML to annotate samples from multiple experiment types" section. The original ELMo paper shows that ELMo embeddings can disambiguate between different definitions of the word "play" based on context, so I wouldn't expect words to have similar embeddings caused by similar spelling. Is it possible that the mouse samples in the training set are frequently annotated as being muscle samples?

Response: We use the pre-trained ELMo model as part of the flair pipeline, which creates word embeddings from characters and not the entire word. Then, word embeddings are created by aggregating character-level information. The reviewer is right in pointing out that ELMo is able to differentiate between different meanings of the same word, it is able to do so only when the final task is integrated with the embedding phase of these neural network models. In our application, however, we use flair (which includes ELMo) only to create word embeddings and later, in a separate step, use these embeddings to design a machine learning model that associates these embeddings to tissue/cell-type terms. Hence, we get one embedding per word and not multiple ones based on different contexts. The only way different words in a sample's description influence each other is during the creation of sample embeddings by taking a weighted average of the individual words in the sample description. To reiterate, this averaging considers all the words together without retaining any context of the words 'surrounding' a given word.

Second, the embedding of each word encodes information from the characters in that word. This allows us to create embeddings for any piece of text without issues that could be posed by misspellings or out of vocabulary words. Therefore, it remains possible that similarly spelled words could be introducing noise into the predictions from our models by ending up with similar word embeddings generated from their similar spellings.

We have now revised the language of the manuscript to reflect these nuances: "The low performance of the muscle model (precision = 0.58) could be explained by the presence of the word "musculus" (indicative of a mouse sample) in the majority of samples incorrectly annotated for muscle tissue. This could be the result of a character-level embedding method like flair yielding similar embeddings for similarly spelled words, which may not be an issue for token-level embedding methods that consider the full word."

We examined the cosine similarity between the embedding for the word "muscle" and the word "musculus" and found the similarity to be 0.43. A value of 0 would indicate no similarity, and a value of 1 would indicate the embeddings are perfectly similar (since embedding dimensions can be negative, the cosine similarity can be -1 for perfect dissimilarity). A value of 0.43, while not exceedingly large, is non-negligible and could support our hypothesis. In regards to the training set, there are no samples in our gold standard where "mouse" or "musculus" are co-mentioned in samples positively annotated for muscle tissue.

Nevertheless, the reviewer brings up a very good point, which is: how does one explain any given prediction by a model like NLP-ML that uses abstract word embedding vectors that are then averaged across tens/hundreds of words within a sample text? Investigating this problem is a worthwhile endeavor and is part of our future research.

Comment 2.3:

I appreciate the authors' use of dataset-aware train/validation set splitting to avoid data leakage and overoptimistic estimates of performance

Response: Thank you!

Comment 2.4:

Table 2 may not be a representative sample of the models' predictions? Most of the models do well for the predictions they're most confident in, but would the results for, e.g., ten random predictions with predicted probability > .9 show the same distribution? This experiment may be infeasible to run due to the manual evaluations required though.

Response: This is a good point and we wish to begin by highlighting an observation about sample descriptions. It is typical for samples in the same experiment to have very similar text descriptions and, based on our manual inspection, the samples with the top predicted probability for each tissue or cell type are unlikely to all come from the same experiment because their descriptions were not similar to each other. It is therefore highly likely that the majority of samples with predicted probabilities > 0.9 in these cases come from a small number of experiments. As a result, sampling 10 random samples from the highest predicted probabilities would most likely lead to samples from the same experiments as the top predicted samples that we manually evaluated, which we feel to be representative. As mentioned by the reviewer, it might be more robust to sample multiple sets of samples with high predicted probabilities, evaluate each set manually, and average the metrics, but the manual curation effort required to conduct that evaluation makes this analysis very challenging. Nevertheless, when a researcher uses our method to identify samples from their tissue/cell-type of interest, the sample ranking will be accompanied by the original sample descriptions, which the researcher can easily examine to make an assessment on a case-by-case basis.

Comment 2.5:

It is unclear why the authors decided on a one-vs-rest formulation for the logistic regression model instead of using multi-class logistic regression.

Response: We agree with the reasoning behind this question: a single multi-class classifier would be ideal. However, we chose one-vs-rest classifiers rather than multi-class logistic regression because it allows us to impose ontological structure into our training step. To elaborate, a multi-class classifier would be sufficient if all the samples in our gold standard were either a positive or a negative for every tissue or cell type (just like, say, an animal image classification task where each image is marked with each animal being present or not in an image). However, that is not the case in our application. For a particular tissue or cell type term, we have 'positive', 'negative', and 'neutral' samples from our gold standard based on the tissues those samples are directly annotated to and the relationship of those tissues to the given term of

interest in the underlying tissue ontology. 'Positive' samples are those that are directly annotated to the term of interest or any of that term's descendents in the ontology. For example, for kidney (UBERON:0002113), samples labeled for kidney would be marked as positives along with samples annotated for terms like kidney cell (CL:1000497), kidney epithelium (UBERON:0004819), or nephric duct (UBERON:0009201), which are all descendant terms of 'kidney' in the UBERON ontology. We then identify all samples that are annotated to the ancestors of the term of interest in the ontology and mark them as 'neutral' because these samples *could* belong to the tissue under consideration. In the instance of kidney, any sample annotated to upper urinary tract (UBERON:0011143), for example, would be a neutral because it is unclear whether it should be labeled a positive or a negative for kidney without further information because either could be true. All other samples in the gold standard are declared as 'negatives'. This label structure, which fully respects our incomplete knowledge and the ontology structure, is difficult to provide as target label vectors within multi-class logistic regression, which expects either a 'positive' or a 'negative' label at each output. Hence, we chose one-vs-rest classifiers.

Reviewer #3 (Remarks to the Author):

Hawkins et al present a method "NLP-ML" to infer tissue and cell type annotations from free text metadata, and compare the performance of their system to two well established approaches, TAGGER and MetaSRA, as well as direct tissue annotation from expression data. Demonstrated performance is better than the other two text-based methods, but slightly less good than the expression database based approach. The authors argue that NLP-ML nevertheless improves on the state of the art, as the method

- helps to improve performance overall in combination of multiple approaches, and
- is relatively easy to apply, and might be useful for the annotation of multiple omics data types, as it relies only on free text metadata, rather than highly structured expression data.

The method is well described and documented, including source code and available/referenced datasets.

* Major concerns:

Comment 3.1:

The authors claim, even in the title, a potential to generalise the method to "omics" samples, which is the major claim to progress beyond state of the art, compared to expression based methods. However, this is only demonstrated based on data from two databases (Geo, ArrayExpress) and five relatively "related" methods. To support this claim of potential to generalise, it would be helpful to apply the method to data from a different database and field. Proteomics would be a potential example, as there are enough public datasets available, and the sample character is still related. Metabolomics would be a more challenging demonstrator, both in terms of data availability and divergence of sample types.

Response: This is a fair point that is also made by Reviewer 3. In principle, our method is general and can be applied to annotate samples from any –omics experiment type beyond genomics/transcriptomics. The reviewer is right that each –omics molecular data presents its own technical challenges. However, as our approach annotates each sample only based on its text description and not based on the recorded molecular data (e.g., expression profile or methylation profile), these technical variations will not affect NLP-ML. If there are systematically different ways in which researchers *describe* samples from different –omics types, that might affect our method, but currently there is no evidence for such systematic textual differences.

Nevertheless, as we have detailed below (*Validation on other omics types*), without a gold standard to evaluate our model predictions against and the infeasible manual curation effort required to validate predictions based on external information (e.g., information hidden away in the papers describing the datasets), it is not possible to unequivocally claim that our method works well across all –omics types.

Therefore, we have revised the wording in our manuscript to now only make claims pertaining to predicting tissues and cell types for *genomics* data rather than *-omics* data in general. The term “-omics” remains in the manuscript in a few areas where the discussion is broader than just the results conveyed from our work, but claims pertaining to our NLP-ML models have been revised.

Validation on other omics types: In an effort to validate our NLP-ML predictions on samples from other omics data types, we examined proteomics data from two databases: PrideDB (<https://www.ebi.ac.uk/pride/>) and OmicsDI (<https://www.omicsdi.org/>), which contain metadata for a large number of proteomics and -omics experiments, respectively. PrideDB not only contains plain text metadata for each proteomics experiment, but in many cases, the experiments also include tissue annotations. These annotations, presumably, are submitted by experimenters with their submission to the database. However, PrideDB does not include any sample-level information. Instead, the database contains sample protocol information, which is more akin to a paragraph from the Methods section of an accompanying manuscript and less indicative of actual sample-level information. We sought to use OmicsDI for doing a broader -omics evaluation, but the proteomics data from OmicsDI is pulled directly from PrideDB, and many larger -omics types lack gold standard labels like we would have working with PrideDB metadata. Evaluating our approach on experiment protocol description – which are more similar to text in Methods sections describing the full experiment and less to descriptive sample metadata – will be unfair. Conducting a fair evaluation entails a substantial manual curation process. Therefore, we have revised the wording of our manuscript to only make claims about genomics samples.

Comment 3.2:

Minor concerns:

Somewhere early on, "available samples" should be defined, it might be interpreted as physical samples available from providers. The subject of this manuscript are "available sample descriptions". On a side note, not a mandatory revision, a really interesting extension of the manuscript might be to map the samples to actually available samples from biobanks etc.

Response: Thank you. In the first line of our Introduction, we now say “data from >1.3 million human –omics samples and >26,000 –omics datasets that are publicly available”. And, yes, connecting samples to actually available samples in biobanks would be valuable and this is the work two large data repositories BioSample (<https://www.ncbi.nlm.nih.gov/biosample/>) and BioSamples (<https://www.ebi.ac.uk/biosamples/>) are attempting to address in a comprehensive manner.

Comment 3.3:

P2: "continues to grow exponentially". Do available samples really grow exponentially?

Response: Yes, omics samples are indeed growing exponentially! We have included the following references to support this statement:

Krassowski, Michal, Vivek Das, Sangram K. Sahu, and Biswapriya B. Misra. 2020. “State of the Field in Multi-Omics Research: From Computational Needs to Data Mining and Sharing.” *Frontiers in Genetics* 11: 1598. <https://doi.org/10.3389/fgene.2020.610798>.

Conesa, Ana, and Stephan Beck. 2019. “Making Multi-Omics Data Accessible to Researchers.” *Scientific Data* 6 (October): 251. <https://doi.org/10.1038/s41597-019-0258-4>.

Perez-Riverol, Yasset, Andrey Zorin, Gaurhari Dass, Manh-Tu Vu, Pan Xu, Mihai Glont, Juan Antonio Vizcaíno, et al. 2019. “Quantifying the Impact of Public Omics Data.” *Nature Communications* 10 (1): 3512. <https://doi.org/10.1038/s41467-019-11461-w>.

Stephens, Zachary D., Skylar Y. Lee, Faraz Faghri, Roy H. Campbell, Chengxiang Zhai, Miles J. Efron, Ravishankar Iyer, Michael C. Schatz, Saurabh Sinha, and Gene E. Robinson. 2015. “Big Data: Astronomical or Genomical?” *PLOS Biology* 13 (7): e1002195. <https://doi.org/10.1371/journal.pbio.1002195>.

Comment 3.4:

P2: ArrayExpress ref might be updated.

Response: We have updated the references to include the following: Sarkans, Ugis, Anja Füllgrabe, Ahmed Ali, Awais Athar, Ehsan Behrangi, Nestor Diaz, Silvie Fexova, et al. 2021. “From ArrayExpress to BioStudies.” *Nucleic Acids Research* 49 (D1): D1502–6. <https://doi.org/10.1093/nar/gkaa1062>.

Comment 3.5:

On pages 3 and 14, the authors mention that MetaSRA is "slow", "low throughput" in comparison to NLP-ML. This should be supported by objective measures.

Response: The text on page 14 has been changed to the following: “Secondly, the method is low throughput, requiring a large amount of time and computational resources to process a single piece of text. The full MetaSRA pipeline needs to be executed for each unique input. For instance, annotating our >11,000 samples meant generating a unique input for each description and running the full computation pipeline for each input individually. The average runtime for

annotating each sample was approximately 1 hour. The average runtime for dataset descriptions often exceeded 3 hours.”

We have also updated a paragraph in Discussion to include additional points related to the comparison between NLP-ML and MetaSRA: “Our NLP-ML method has a number of specific advantages compared to existing text and expression-based solutions to annotating samples for tissues and cell types. Our models are able to make predictions for any genomics sample given a plain-text sample description, which lends itself to predictive flexibility compared to methods that use the underlying molecular data to make tissue or cell type annotations. These descriptions can be any unstructured plain text. This is a key advantage over MetaSRA, which was designed for leveraging structured key-value data (particularly the ‘Characteristics’ field) in order to construct knowledge graphs for annotating samples. NLP-ML is also computationally lightweight: predictions for >300 fully trained models can be made on dozens of pieces of text in a matter of minutes on a modest local computer. This is significantly faster than MetaSRA, which takes on the order of hours for sample descriptions and needs to be executed for each individual piece of text, and Tagger, which needs to load large dictionaries into memory before doing an exhaustive, exact-string matching to the dictionary. MetaSRA was especially designed to operate on very small pieces of text (key-value pairs). Our method outperforms other text-based methods while maintaining biological interpretability both in terms of how the models are trained (taking into account ontology structure when assigning training labels) and in how the models perform (Figure 4 and 5), which when combined with the other benefits of NLP-ML – predominantly scalability, efficiency, and the ability to work on unstructured text from any source – set it apart from existing text-based methods. Further, because NLP-ML addresses a more general problem, *i.e.*, annotating large collections of unstructured text, it can easily be applied to any text data including descriptions of more –omics data types beyond gene expression.”

Comment 3.6:

P3, line 3: "...several false positives". Several is an odd quantification here.

Response: This language has been changed: “However, without an additional step of manual curation, NER-based methods suffer from high false-positive rates due to the presence of varied and conflicting pieces of information in sample descriptions.”

Comment 3.7:

P4: Add literature reference for Ontology Lookup Service.

Response: Thank you. We added the following reference for OLS: Jupp, S., T. Burdett, C. Leroy, and H. Parkinson. 2015. “A New Ontology Lookup Service at EMBL-EBI.” In *SWAT4LS*. This is one of the first works to outline the platform in detail. In combination with the URL for OLS, we feel this is sufficiently cited.

Comment 3.8:

P18: "perform similarly overall" is a bit of an idealising statement.

Response: The language has been changed to “the performances of expression-based models and NLP-ML models are comparable in terms of overall performances.” This statement is based on the comparison of the distributions of model performances between expression-based predictions and NLP-ML predictions, which results in a corrected Wilcoxon rank sum test p-value of 0.12, indicating that there is not a statistically significant difference between their overall auPRC values.

Comment 3.9:

Fig 6, panel A, top row; fig S10, panel B, top row: The boxplots show a strange artifact at the right border, exceeding "1". While probably a problem of the underlying library, it would be nice to correct this for a potential next version.

Response: The area of the plot that exceeds 1 is a statistical artifact from how the notch on the boxplot is calculated and not a reflection of the true data. The notch is a visual indicator of the confidence interval around the median, which is calculated as follows: $median \pm \frac{IQR}{\sqrt{n}}$, where IQR is the interquartile range and n is the sample size of the boxplot. Depending on the values of the IQR and n , it is possible for the notch to exceed the maximum auPRC of 1, and should be treated as a piece of visual information about the estimate of the median rather than a true datapoint. The statements made in the paper are based on robust statistical tests.

REVIEWER COMMENTS

Reviewer #1 (Remarks to the Author):

Comments for Author:

The efforts made by Hawkins et al. in their revised manuscript are appreciated, as these updates address some of the primary concerns with the original submission. The primary research goal remains sound; an unification of NLP methods with ontology-driven metadata standardization has the potential to offer noticeable benefits to all researchers seeking to use and re-use omics data.

However, the manuscript itself remains challenged on providing specific details regarding the domain that NLP-ML (and txt2onto) has been tested on (i.e., for “genomics” vs. all omics); this is not a small but rather crucial concern. Several essential aspects of the manuscript also remain inadequately addressed. All of these severely limit the potential impact of the described methods. Though the authors have claimed that their method can handle variations in sample type within and across data sets, they offered limited evidence to support their speculation, at least partially due to a dearth of sufficiently labeled training data. Another absolutely essential detail remains missing: is there a specific use case where these advantages apply? Is slow software causing a bottleneck in omics metadata creation? Is a lack of flexibility serving to limit the number of accurately annotated data sets? The answer to all of these questions may be a resounding ‘yes’, but the authors must clearly identify why and how. These and additional concerns are elaborated upon below.

This proposed approach is new but supporting evidence remains very preliminary. For this reason, adequately detailed use cases with demonstrated positive improvement/benefits are essential.

Issues/Comments:

(1) The inclusion of further details regarding advantages as compared to MetaSRA are helpful for providing justification and context. These improvements are considerable, and should certainly be acknowledged, not the least because of the clear improvement in application flexibility provided by NLP on free text vs. small key-value pairs. The expected broader impact on those downstream applications remains unclear. As the authors clearly delineate in their introduction, well over a million omics samples exist, but is NLP-ML an effective solution for considerably improving metadata amongst these datasets?

(2) The authors’ response to concerns regarding false positives in NER is informative. More of the details regarding this analysis should be provided in the manuscript itself.

(3) Regarding the replacement of “-omics” with “genomics”, this appears to be a reasonable change reflecting the extent to which the current methods may be tested with any degree of confidence. The lack of sample-level information in proteomics databases is an open challenge and certainly limits the extent to which proteomics metadata may be extended. This may be a case in which careful human annotation may be effective (yet, as is often the case, demanding of considerable time and labor). The authors also state in their response that “...there is no evidence for such systematic textual differences” in descriptions of different omics types. This is a difficult claim to support without a direct comparison of the vocabulary of different omics metadata.

(4) Availability of code to train NLP-ML models is appreciated. While creation of a more comprehensive resource may extend beyond the scope of the manuscript, the requested online demonstration was more along the lines of that provided for MetaSRA at <https://metasra.biostat.wisc.edu/>, or a publicly available demonstration allowing advantages of NLP-ML to be shown without the necessary programming skill or environment setup prerequisites. In practice, even a Docker image may be sufficient.

Reviewer #2 (Remarks to the Author):

All of my comments have been addressed by the authors.

Reviewer #3 (Remarks to the Author):

The authors have appropriately addressed my concerns, with the exception of 3.9, the inaccuracy in figure S10. I guess it's now an editorial decision on if/how this should be fixed, I'd just crop away the part beyond 1 with a standard graphics tool.

Reviewer #1 (Remarks to the Author)

Comment: The efforts made by Hawkins et al. in their revised manuscript are appreciated, as these updates address some of the primary concerns with the original submission. The primary research goal remains sound; an unification of NLP methods with ontology-driven metadata standardization has the potential to offer noticeable benefits to all researchers seeking to use and re-use omics data.

However, the manuscript itself remains challenged on providing specific details regarding the domain that NLP-ML (and txt2onto) has been tested on (i.e., for “genomics” vs. all omics); this is not a small but rather crucial concern. Several essential aspects of the manuscript also remain inadequately addressed. All of these severely limit the potential impact of the described methods. Though the authors have claimed that their method can handle variations in sample type within and across data sets, they offered limited evidence to support their speculation, at least partially due to a dearth of sufficiently labeled training data. Another absolutely essential detail remains missing: is there a specific use case where these advantages apply? Is slow software causing a bottleneck in omics metadata creation? Is a lack of flexibility serving to limit the number of accurately annotated data sets? The answer to all of these questions may be a resounding ‘yes’, but the authors must clearly identify why and how. These and additional concerns are elaborated upon below.

This proposed approach is new but supporting evidence remains very preliminary. For this reason, adequately detailed use cases with demonstrated positive improvement/benefits are essential.

Response:

Domain that NLP-ML has been tested on:

We have already made this change in our previous resubmission of the manuscript, amending our claims to pertain only to *genomics* data and not *-omics* data as a whole. Hence, now, the term “-omics” is mentioned only in the sections of the manuscript where general discussions are appropriate. All claims based on our analyses and results are restricted to making predictions on “genomics” data alone. To make the domain of our application clearer, we

have also added a paragraph in the *Discussion* section that outlines how our approach (or any other approach for sample annotation) cannot be used on other –omics data such as proteomics (from databases like PRIDE) due to the lack of sample-level information. Therefore, together, we believe that this concern has been addressed.

Cases where advantages apply:

Substantially better accuracy: The single biggest advantage of our approach is its substantial increase in sample annotation accuracy over and above approaches such as MetaSRA. We have demonstrated this advantage throughout the manuscript based on: **i)** rigorous evaluation based on a high-quality gold standard (**Figures 2, 3, 6, S1–8, and S10–12**), **ii)** application to completely independent samples from other genomics samples not part of the gold standard (**Table 2**), and **iii)** Manual inspection of several sample and dataset descriptions that highlights specific cases where our approach mitigates false positives and false negatives (compared to existing methods) to result in accurate sample annotations (**Supp. Notes 1 and 2**). Together, the overall improvement in sample annotation accuracy makes our approach broadly usable to annotate samples across the board.

Flexibility: The second major advantage of our approach is its flexibility, that is, its ability to take as input any unstructured sample description without requiring structured key-value pairs. This is a well-recognized problem discussed in many other studies (cited in our manuscript, including [i] Byrd *et. al.*, (2020) “Responsible, Practical Genomic Data Sharing That Accelerates Research.” *Nature Reviews Genetics*; and [ii] Rajesh *et. al.*, (2021) “Improving the Completeness of Public Metadata Accompanying Omics Studies.” *Genome Biology*). We have also highlighted several examples in **Supp. Notes 1 and 2** where samples lack structured sample descriptions. Therefore, the flexibility of our approach makes it applicable to all genomics samples irrespective of the format of their text descriptions.

Software speed: High accuracy, flexibility, and generalizable software design makes our approach perfect for incorporation into existing computational sample annotation workflows that bioinformaticians, computational biologists, and data analysts will run on hundreds/thousands of sample descriptions. The substantial software speed (few seconds per sample/dataset for our method compared to 1–3 hours for MetaSRA) is a major advantage both in the context of re-annotating millions of existing samples and for annotating the exponentially growing number of new samples that are being deposited in public databases. This high speed is *even more vital* in a futuristic setting where our NLP-ML models are used on-the-fly to suggest structured annotations to samples as-and-when their descriptions are being entered into online forms by data submitters.

Comment 1.1: The inclusion of further details regarding advantages as compared to MetaSRA are helpful for providing justification and context. These improvements are considerable, and should certainly be acknowledged, not the least because of the clear improvement in application flexibility provided by NLP on free text vs. small key-value pairs. The expected broader impact on those downstream applications remains unclear. As the authors clearly delineate in their introduction, well over a million omics samples exist, but is NLP-ML an effective solution for considerably improving metadata amongst these datasets?

Response 1.1: NLP-ML is an effective solution for considerably improving metadata amongst millions of omics samples for the following reasons:

1) *Substantially better accuracy:* The single biggest advantage of our approach is its substantial increase in sample annotation accuracy over and above approaches such as MetaSRA. We have demonstrated this advantage throughout the manuscript based on: **i)** rigorous evaluation based on a high-quality gold standard (**Figures 2, 3, 6, S1–8, and S10–12**), **ii)** application to completely independent samples from other genomics samples not part of the gold standard (**Table 2**), and **iii)** Manual inspection of several sample and dataset descriptions that highlights specific cases where our approach mitigates false positives and false negatives (compared to existing methods) to result in accurate sample annotations (**Supp. Notes 1 and 2**). Together, the overall improvement in sample annotation accuracy makes our approach broadly usable to annotate samples across the board.

2) *Flexibility:* The second major advantage of our approach is its flexibility, that is, its ability to take as input any unstructured sample description without requiring structured key-value pairs. This is a well-recognized problem discussed in many other studies (cited in our manuscript, including [i] Byrd *et. al.*, (2020) “Responsible, Practical Genomic Data Sharing That Accelerates Research.” *Nature Reviews Genetics*; and [ii] Rajesh *et. al.*, (2021) “Improving the Completeness of Public Metadata Accompanying Omics Studies.” *Genome Biology*). We have also highlighted several examples in **Supp. Notes 1 and 2** where samples lack structured sample descriptions. Therefore, the flexibility of our approach makes it applicable to all genomics samples irrespective of the format of their text descriptions.

2) *Software speed:* High accuracy, flexibility, and generalizable software design makes our approach perfect for incorporation into existing computational sample annotation workflows that bioinformaticians, computational biologists, and data analysts will run on hundreds/thousands of sample descriptions. The substantial software speed (few seconds per sample/dataset for our method compared to 1–3 hours for MetaSRA) is a major advantage both in the context of re-annotating millions of existing samples and for annotating the exponentially growing number of new samples that are being deposited in public databases.

This high speed is *even more vital* in a futuristic setting where our NLP-ML models are used on-the-fly to suggest structured annotations to samples as-and-when their descriptions are being entered into online forms by data submitters.

The immediate expected broad impact of our approach will be through its incorporation into existing computational sample annotation workflows that bioinformatics and computational data analysts will run on hundreds/thousands of sample descriptions. Therefore, to ensure that our approach can be easily used by computational biologists, we have released a well-documented Python software (txt2onto) and have provided two use cases on our github repository: i) to train custom text-based machine learning models using our approach (to predict any sample attribute based on the sample description), and ii) to apply the trained model to predict the desired sample annotations on a large number of new samples.

The way in which our approach empowers new discoveries is by providing structured annotations to publicly available genomics samples so that biologists can easily find the samples (and datasets) relevant to their question of interest from the ocean of hundreds of thousands of samples. Without the annotations provided by our approach, searching based only

on the original sample description text will lead to numerous false positives and false negatives. Even with annotations provided by a competitive method like MetaSRA (compared to in this study), biologists will find several false positive samples (i.e., samples not really related to the tissue/cell-type of their interest). Using the annotations provided by our approach will enable researchers to more accurately find the samples they want.

We have highlighted many such cases of samples correctly identified by our approach (overcoming false positive and false negative predictions by other methods) in the Supplemental Notes 1 and 2.

Once biologists have found their samples of interest, the analyses they subsequently carry out may lead to novel discoveries of various forms. Thus, ultimately, our approach helps democratize data-driven biology by enabling biologists to easily discover publicly available genomics data.

Comment 1.2: The authors' response to concerns regarding false positives in NER is informative. More of the details regarding this analysis should be provided in the manuscript itself.

Response 1.2: As per the suggestion of the reviewer, the following text has been included in the manuscript:

Examining specific sample descriptions (Supplemental Note 2) also showcases how NLP-ML is able to achieve lower false positive rates by taking advantage of the overrepresentation of the true tissue name (compared to mentions of other, non-source tissues or cell types) in the descriptions.

We have also included the following text (from our previous reviewer responses) as Supplemental Note 2:

To elucidate our method's behavior for controlling for false positives over NER, we examined all of the cases where for a given sample and for a particular tissue or cell type, the true label according to our gold standard is negative, and our method (NLP-ML) correctly labels the sample as such (true negative) but either MetaSRA or Tagger labels the sample as a positive (false positive). We then filtered these instances to ones where the predicted probability from NLP-ML is < 0.05 to examine cases where our models were confident in assigning a negative label, and further filtered these cases to instances pertaining to a tissue or cell types whose auPRC from cross validation is > 0.80 to only consider tissues and cell types where the predicted probabilities from NLP-ML are most likely to be accurate. Below, we describe our observations from three specific tissues or cell types along with a count of the number of samples that fulfilled the above criteria.

For brain (N = 12), for all of the cases where NLP-ML correctly identified a non-brain sample correctly as a negative but the other text-based methods did not, the samples in question came from liver or blood, but all came from either patients who are brain dead or patients with brain cancer. For liver (N = 26), the true label for the samples were either blood or colon (specifically samples from colon adenocarcinoma tumors), but the patients were either liver transplant patients in the case of the true label being blood, or the word "liver" just appears

in the sample description. For intestine (N = 23), all samples were from stomach stromal tumor, but terms like “gastrointestinal” and “small intestine” are mentioned throughout.

These instances point to one hypothesis about how NLP-ML might have been able to correctly label these samples as negatives for the appropriate tissues. In almost all cases where NLP-ML correctly predicts a negative and the other text-based methods incorrectly assign a positive label, the true label tissue name appears more times than any other tissues or cell types in the description. This hypothesis is supported by how we generate a text-based feature vector for a sample based on its ‘bag-of-words’ (from the description) where more frequently appearing words directly contribute more strongly to the final feature vector, making it more associated with the correct tissue name and less associated with the incorrect ones. We also suspect that there may be some words present in the description’s bag of words that provide additional contextual clues that can additionally point NLP-ML to the true tissue of origin, thus contributing to NLP-ML’s lower false positive rate.”

Comment 1.3a: Regarding the replacement of “-omics” with “genomics”, this appears to be a reasonable change reflecting the extent to which the current methods may be tested with any degree of confidence. The lack of sample-level information in proteomics databases is an open challenge and certainly limits the extent to which proteomics metadata may be extended. This may be a case in which careful human annotation may be effective (yet, as is often the case, demanding of considerable time and labor).

Response 1.3a: We completely agree. We have now explicitly noted this point in our manuscript as follows:

Further, because NLP-ML addresses a more general problem, i.e., annotating large collections of unstructured text, it can easily be applied to any text data including descriptions of more – omics data types beyond genomics. However, a significant challenge faced by the biomedical community – an open challenge recognized by funding agencies such as NIH and data consortia such as NCI Data Commons – is the lack of sample-level information that methods like ours can utilize. For instance, databases such as PRIDE (Perez-Riverol 2019), the preeminent public database for proteomics datasets, only includes descriptions of entire datasets and experiments and not of individual samples. Addressing this challenge of lack of sample-level descriptions requires careful human annotation using semi-automated systems such as ZOOMA (<https://www.ebi.ac.uk/spot/zooma/>) based on descriptions about samples available elsewhere, including accompanying publications.

Comment 1.3b: The authors also state in their response that “...there is no evidence for such systematic textual differences” in descriptions of different omics types. This is a difficult claim to support without a direct comparison of the vocabulary of different omics metadata.

Response 1.3b: We would like to clarify the language of our previous response on this issue. Since our approach annotates each sample only based on its unstructured text description and not based on the recorded molecular data (e.g., expression profile or methylation profile), these technical variations will not affect NLP-ML.

Based on our experience and empirical observations, we *expect* that biomedical researchers, in general, tend to describe biological samples (*i.e.* attributes about their source such as tissue, cell type, phenotype, environment, and treatment) in a similar manner regardless of the type of –omics data being generated from those samples (*e.g.*, gene expression *vs.* methylation *vs.* proteomics).

Ascertaining this expectation one way or the other is hard due to the limited availability of sample-level information across –omics types (*e.g.*, proteomics). A conclusive vocabulary-based analysis is, unfortunately, not feasible without substantial manual effort. We may undertake such an effort as part of our future work on extending our NLP-ML method to other –omics types, in which case we will establish any systematic differences in the vocabulary of sample-level information.

Comment 1.4: Availability of code to train NLP-ML models is appreciated. While creation of a more comprehensive resource may extend beyond the scope of the manuscript, the requested online demonstration was more along the lines of that provided for MetaSRA at <https://metasra.biostat.wisc.edu/>, or a publicly available demonstration allowing advantages of NLP-ML to be shown without the necessary programming skill or environment setup prerequisites. In practice, even a Docker image may be sufficient.

Response 1.4: Our method was designed with computational text annotation workflows in mind. By that, we mean that existing computational pipelines that aim to classify sample text descriptions (or any biomedical text) can readily use our source code to both train custom NLP-ML models and predict based on our pre-trained or newly-built custom models. Supplementing existing workflows this way will enable making predictions on hundreds of thousands of samples.

Developing a solution that enables the use of NLP-ML without the necessary programming skill or environment setup is a valuable one. The issue here is that there is a substantial skill gap between a computational biologist wanting to use our software (txt2onto) and an experimental biologist wishing to retrieve relevant samples based on annotations by NLP-ML. Based on our experience, a containerized solution such as a Docker image falls somewhere in the middle of this gap because containerized solutions still require a significant amount of overhead to setup and execute, which in turn definitely requires knowledge of the unix command-line. As the reviewer rightly points out, scientists without command-line or programming background are more likely to benefit from a queryable, interactive webserver with NLP-ML as the predictive backbone that allows enables them to query for a particular tissue/cell-type by typing in keywords in a text box and retrieve predictions (*i.e.* relevant samples and datasets) as output. Developing such a webserver is part of our future work and will allow users to explore the predictions of NLP-ML on thousands of publicly available genomics samples instantly without needing to execute any code in a programming environment.

Reviewer #2 (Remarks to the Author):

Comment 2.1: All of my comments have been addressed by the authors.

Response 2.1: Thank you!

Reviewer #3 (Remarks to the Author):

Comment 3.1: The authors have appropriately addressed my concerns, with the exception of 3.9, the inaccuracy in figure S10. I guess it's now an editorial decision on if/how this should be fixed, I'd just crop away the part beyond 1 with a standard graphics tool.

Response 3.1: We appreciate the feedback of the reviewer on this matter. We would like to clarify that the aspect of Figure S10 in question is not an “inaccuracy,” but rather an explainable artifact of a well-established method for calculating the “notch” of a boxplot based on data to enable visual inference of significant differences between boxplots. For example, see <https://www.rdocumentation.org/packages/grDevices/versions/3.6.2/topics/boxplot.stats>, under section “Details”:

“The notches (if requested) extend to $\pm 1.58 \text{ IQR}/\sqrt{n}$. This seems to be based on the same calculations as the formula with 1.57 in Chambers et al (1983, p.62), given in McGill et al (1978, p.16). They are based on asymptotic normality of the median and roughly equal sample sizes for the two medians being compared, and are said to be rather insensitive to the underlying distributions of the samples. The idea appears to be to give roughly a 95% confidence interval for the difference in two medians.”

Nevertheless, to ensure that readers get the accurate picture of the underlying data, as part of the GitHub repository included with the manuscript (<https://github.com/krishnanlab/txt2onto>), we have included files containing the data underlying all our plots – including Figure S10 https://github.com/krishnanlab/txt2onto/blob/main/paper_results/ExpressionComparisonsWithNLPPerformances.csv. This file shows that none of the auRPC values exceed 1.0. Hence, we believe that changes to Figure S10 are not necessary.